# *Toxoplasma gondii* infection of neurons alters the production and content of extracellular vesicles directing astrocyte phenotype and contributing to the loss of GLT-1 in the infected brain

**Emily Z. Tabaie[1], Ziting Gao[2], Nala Kachour[1], Arzu Ulu[1], Stacey Gomez[1], Zoe A. Figueroa[1], Kristina V. Bergersen[1], Wenwan Zhong[2], Emma H. Wilson**[1]*

**1** Division of Biomedical Sciences, School of Medicine, University of California, Riverside, California, United States of America, **2** Department of Chemistry, University of California, Riverside, California, United States of America

* emmaw@ucr.edu

## Abstract

*Toxoplasma gondii (T. gondii),* a prolific protozoan parasite, forms cysts within neurons of the central nervous system that maintain infection for the lifetime of the host. Astrocytes are fundamental to neuronal health by providing nutrients and structural support and help regulate neurotransmitters by continuous communication with neurons. It is not yet known how infection and the presence of intracellular cysts, disrupts the crucial relationship between these cells. Extracellular vesicles (EVs) function in intracellular communication and can contain proteins, lipids, DNA, miRNA, and other RNA subtypes. EVs are produced by all cells and play an important role in neuronal-astrocyte interactions, including the regulation of glutamate receptors on astrocytes. Previous work has demonstrated that Toxoplasma infection reduces astrocytic expression of the primary glutamate transporter, GLT-1. Here we tested if cyst infection of neurons alters the production and content of EVs. EVs were isolated from uninfected and infected primary murine cortical neurons and their size, concentration, and characterization were confirmed with nanoparticle tracking analysis (NTA), transmission electron microscopy (TEM), ELISA, western blot, liquid chromatography (LC)-mass spectrometry (MS)/MS, and microRNA sequencing. Analysis reveals that infection of neurons reduced neuronal production of EVs and altered their protein and miRNA content. In addition to changes in host protein content, EVs from infected neurons contained the Toxoplasma proteins GRA1, GRA2, GRA7, MAG1 and MAG2. Following incubation of neuronal EVs with primary astrocytes, GRA7 protein could be observed within intracellular EVs and the nuclei of GRA7+ EV-containing cells. EVs from infected neurons altered gene expression of astrocytes resulting in an increase in pro-inflammatory transcriptional signatures,

**Data availability statement:** The authors declare that all data supporting the findings of this study are available within the article and its supplementary information files. Raw sequencing and processing files for RNA and miRNA Sequencing have been deposited in the GEO database (https://www.ncbi.nlm.nih.gov/geo/) under accession code: PRJNA1158965 (GSE278553 and GSE278754). Raw and processed files for LC-MS/MS protein analysis have been deposited in the MassIVE database (https://massive.ucsd.edu/ProteoSAFe/static/massive.jsp) under accession code: MSV000096222 Some of the original protocols have been published under protocols.io, and the links can be found under the appropriate methodology.

**Funding:** This work was supported, in part, by funding from the National Institute of Health NS137603 and AI158417 to EHW and funds from the Graduate Division at the University of California, Riverside to EHW and from the Division of Biomedical Sciences, UCR to EZT. The funders had no role in study design, data collection and analysis, decision to publish, or preparation of the manuscript.

along with a downregulation of GLT-1 protein expression with similar transcriptional changes found in astrocytes *in vivo*. These results demonstrate the ability of a parasitic infection in the brain to alter EV production and the fundamental communication between neurons and astrocytes.

## Author summary

Infection with the obligate intracellular parasite, *Toxoplasma gondii,* leads to neuronal cysts in the brain for the lifetime of the host. Our lab has previously determined that chronic infection leads to loss of astrocytic glutamate transported, GLT-1, leading to neuronal excitotoxicity. GLT-1 can be regulated by neuronal derived extracellular vesicles (EVs). We wanted to determine if cyst infection of neurons altered EV production and content and if EVs derived from cyst-containing neurons changed astrocyte function. Our study found that Toxoplasma cyst infection decreased EV production by neurons and altered EV host protein and miRNA content. In addition, EVs from infected neurons contained parasite derived proteins including the secreted dense granule protein GRA7. Incubation of these EVs with astrocytes led to EV uptake, GRA7 localization to the nucleus, a decrease in GLT-1 expression, and changes in the transcriptional signature of astrocytes to a pro-inflammatory response. Finally, these changes in astrocytic gene expression could be seen *in vivo* following infection using scRNAseq. This study demonstrates that Toxoplasma cysts alter neuron-astrocyte communication bypassing traditional immune mechanisms of recognition and leading to changes in astrocyte function.

## 1. Introduction

Nearly one-third of the world's population is infected with the obligate intracellular parasite, *Toxoplasma gondii (T. gondii)* [1,2]. Following a period of rapid replication and systemic inflammation, infection leads to slow growing tissue cysts in the brain, skeletal and cardiac muscle of the host [3]. Along with the transition to the less cell lytic cyst form, chronic infection is mediated by the ability of the parasite to manipulate cell signaling pathways and inhibit host cell defense mechanisms [4] in macrophages [5,6], dendritic cells [7], and neurons [8].

   In the murine brain, cysts vary in size between 5 and 70 µm, are intracellular, and typically reside within neurons. Neurons are responsible for receiving and sending sensory and motor signals, allowing for information to be transmitted across great distances [9]. Although Toxoplasma infection does not cause significant clinical pathology in the immune competent individual, there is evidence cyst infected neurons are functionally compromised with a reduction in dendritic spines and electrical activity [3,10]. The function of neurons is intimately linked with astrocytes, glial central nervous system (CNS) resident cells. Astrocytes are the most

abundant glial cell in the brain and assist with support and nourishment of neurons, promoting signaling, and taking-up nutrients from the blood to transfer to neurons [11]. In addition, astrocytes are vital for maintaining the blood brain barrier (BBB) and their dysfunction can lead to neurological disorders, including neurodegenerative diseases, stroke, and epilepsy [12]. Importantly, astrocyte-to-neuron communication is mediated through multiple channels including direct cell-to-cell contact, secretion of neurotransmitters, hormones and neuromodulators, or the production, secretion and uptake of extracellular vesicles (EVs). Neurons can communicate information to other cells through electrical and chemical synapses, however, recently their ability to transfer material in EVs has established an important additional paracrine cell-cell mechanism. The communication between neurons and astrocytes is vital for various functions, including synapse formation, ion homeostasis, and the regulation of important neurotransmitters [13].

Glutamate is the primary excitatory neurotransmitter in the brain. Astrocytic glutamate transporter, GLT-1, is responsible for 90% of extracellular glutamate uptake in the CNS, a crucial function to prevent glutamate buildup in the synaptic cleft [14]. GLT-1 dysregulation results in excess glutamate concentrations resulting in an excitotoxic state that can lead to spontaneous seizure activity [15,16]. Astrocytes regulate CNS glutamate by adjusting uptake, release, and synthesis into glutamine. We have previously demonstrated that following Toxoplasma infection, there is a decrease in GLT-1 expression resulting in an increase in extracellular glutamate concentrations [10]. GLT-1 expression is dysregulated in many neurological diseases, making its downregulation an important area of research. The exact mechanism of GLT-1 regulation has yet to be identified, however post-translational modifications and extracellular vesicles are at least partially responsible for its regulation [17–19].

EVs are membrane bound molecules that contain proteins, lipids, DNA, miRNA, and other RNA subtypes. All cells are able to secrete EVs and they can be observed in biological fluids including blood, tear, saliva, urine, breast milk, and cerebrospinal fluid [20,21]. Parasites, including *T. gondii*, also have the capability of producing their own EVs [22,23]. EVs are derived from the fusion of multivesicular bodies with the plasma membrane and upon the extracellular release of the intraluminal vesicles, the EVs can dock to the plasma membrane of the target cell and enter through fusion or endocytosis mechanisms [24]. EVs release their content and alter signaling pathways to assist with cellular communication, through an unknown process. One main characteristic of EVs is that they can function in intercellular communication without the need for cell-to-cell contact or specific receptor interactions.

Here we describe the ability of Toxoplasma infection to alter the production and content of EVs from infected neurons and the resulting effect on recipient astrocytes. Results demonstrate a significant decrease in the concentration of EVs from *T. gondii* infected neurons and an alteration in EV protein and miRNA content. EVs from infected neurons altered astrocytes, including an upregulation in inflammatory genes and a significant decrease in astrocytic GLT-1 protein expression. These processes reveal a previously unknown mechanism of parasite-induced changes in the brain that have the potential to broadly alter neurochemistry in the infected CNS.

## 2. Results

### 2.1. *T. gondii* infection of neurons decreases the production of EVs

To determine if infection of neurons alters EV production, EVs were isolated from primary cortical neurons and infected with *T. gondii.* Toxoplasma spontaneously forms cysts in neurons [25–27], however we have found that too many parasites cause rapid and complete neuronal cell death. To determine the maximum number of infected neurons with the least amount of neuronal damage, four different multiplicity of infections (MOIs) (0.25, 0.50, 0.75, and 1.0) were tested. Primary cortical neurons were infected with their respected MOIs, 8 days after culture to ensure full development and confluency of neurons pre-infection, and images were taken 3 days post infection (d.p.i). From $1\times10^5$ seeded embryonic cortical stem cells ~10,000 cells develop fully into neurons and this cell count remains constant irrespective of infection. Neuronal morphology and health were confirmed with MAP2 (microtubule associated protein 2) staining of dendrites, and NeuN

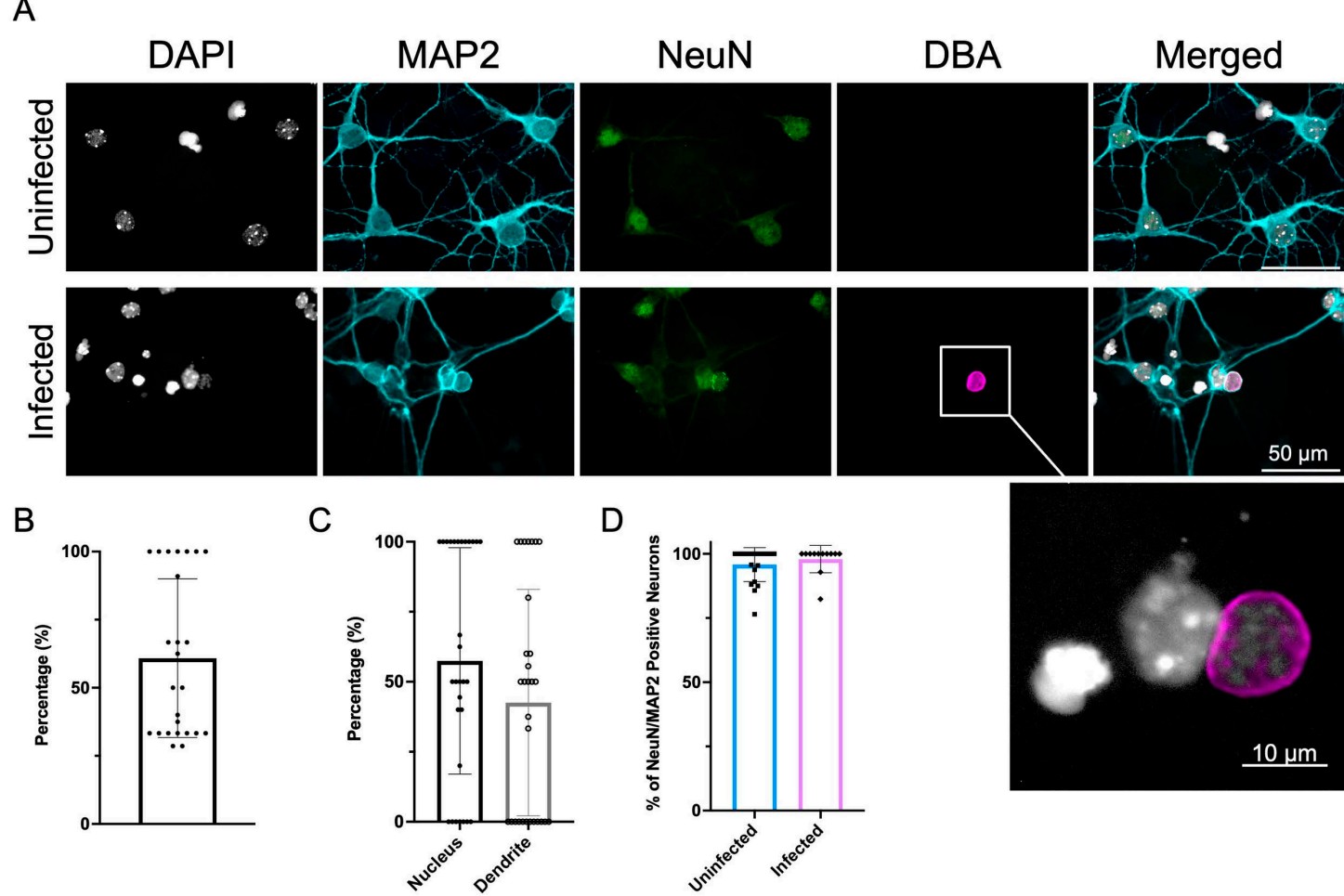

**Fig 1. Primary cortical neurons infected with Toxoplasma.** Infection with a MOI (multiplicity of infection) of 0.50. For quantifications "n" represents a single cell throughout multiple experiments. **(A)** Images taken three days post infection. DAPI stains the nucleus, MAP2 stains the perikarya and dendrites on neurons, NeuN stains the nucleus of the neurons, DBA stains the sugars of the cyst wall. In the infected neurons there is overlap of the cyst and neuron. Scale bar at 50 μm. **(B)** Quantification of the percentage of infected neurons. The number of cysts and neuronal nuclei were counted in order to calculate the percentage of infection. Around half of the neurons were infected with *T. gondii*, which is consistent with the MOI of 0.50. **(C)** Quantification of the percentage of cysts located in the nucleus versus the dendrite of the neuron (Unpaired t-test, n (Nucleus) = 30, n (Dendrite) = 30, Nucleus vs. Dendrite p value = 0.1586). **(D)** Quantification of the percentage of MAP2 and NeuN positive neurons (Unpaired t-test, n (Uninfected) = 21, n (Infected) = 12, Uninfected vs. Infected p value = 0.4533).

(neuronal nuclear antigen) staining of the nucleus (Fig 1A). Cyst generation was confirmed with DBA (*Dolichos biflorus* agglutinin) that binds to sugars of the cyst wall and were found both in the dendrite and soma [3,8,28,29] (Figs 1 and S1). A decrease in dendrite number was observed as MOI increased [30,31] (S1C Fig), consistent with dendritic loss seen *in vivo* following Toxoplasma infection in mice [3,10]. An MOI of 0.50 resulted in >95% of neurons positive for NeuN and MAP2, indicating minimal neuronal toxicity [28,29] (Fig 1D). To maximize the number of infected neurons in culture while optimizing neuronal health, an MOI of 0.50 was chosen for the remaining experiments.

EVs were harvested from neurons at day 3 post infection. To confirm EV characteristics, transmission electron microscopy (TEM) was conducted to determine size and morphological changes upon infection. TEM confirmed the circular morphology of EVs from both uninfected and infected neurons, consistent with previous publications [32–35] (Fig 2A). Quantification of

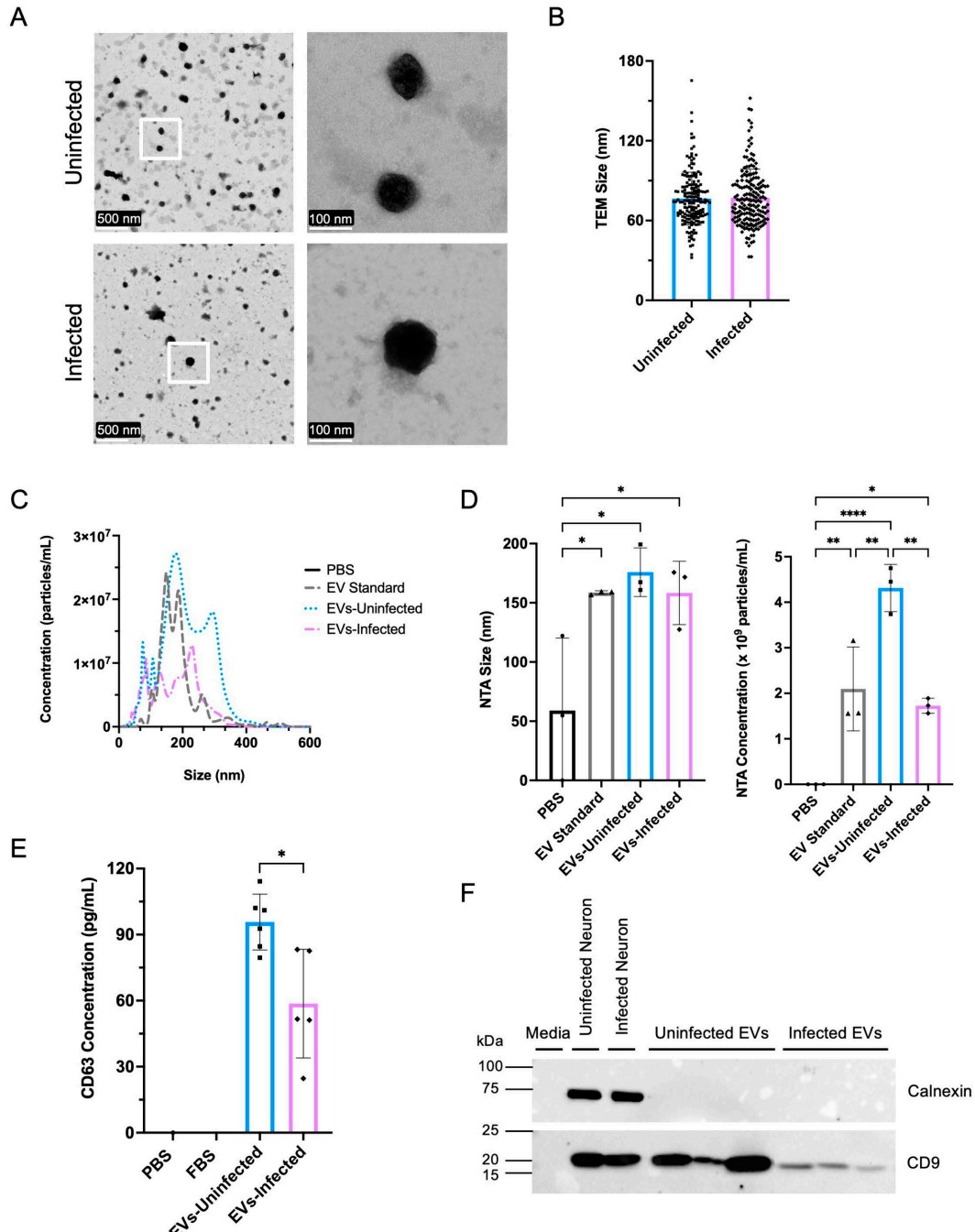

**Fig 2. Toxoplasma infection leads to a decrease in the concentration of EVs. (A)** Transmission Electron Microscopy (TEM) of EVs isolated from uninfected and infected neurons. Right hand panels are zoomed in images of areas outlined by the white box in the corresponding left panels. **(B)** Quantification of the size of EVs from TEM images. "n" represents an EV that was counted throughout multiple slides (Unpaired t-test, n (Uninfected) = 169, n (Infected) = 179, Uninfected vs. Infected p value = 0.7691). **(C)** Nanoparticle Tracking Analysis (NTA) of particles from PBS, EV Standard, EVs isolated from uninfected neurons, and EVs isolated from infected neurons. **(D)** Particle size and concentration of EVs from NTA. "n" represents an EV batch that was run on the NTA or ELISA throughout multiple isolation experiments. Size (One-way ANOVA, n = 3, EV Standard vs. EVs-Uninfected p value = 0.9283, EV Standard vs. EVs-Infected p value = >0.9999, PBS vs. EVs-Uninfected p value = 0.0146, PBS vs. EVs-Infected p value = 0.0338, PBS vs. EV Standard p value = 0.0332, EVs-Uninfected vs. EVs-Infected p value = 0.9245). Concentration (One-way ANOVA, n = 3, EV Standard vs. EVs-Uninfected p value = 0.0042, EV Standard vs. EVs-Infected p value = 0.8309, PBS vs. EVs-Uninfected p value = <0.0001, PBS vs. EVs-Infected p value = 0.0177, PBS

vs. EV Standard p value = 0.0059, EVs-Uninfected vs. EVs-Infected p value = 0.0016). **(E)** Concentration of EVs in neuronal supernatants as determined by CD63 ELISA after centrifugation and ExoEasy Kit (Unpaired t-test, n (PBS) = 1, n (FBS) = 0, n (EVs-Uninfected) = 6, n (EVs-Infected) = 5, EVs-Uninfected vs. EVs-Infected p value = 0.0104). **(F)** Western Blot analysis of EV proteins from uninfected and infected neurons. Infection does not alter EV protein content. Isolated EVs from uninfected and infected neurons containing CD9 (~25 kDa) exposed for 162.3 seconds and not containing Calnexin (~80 kDa) exposed for 40.7 seconds.

EV size demonstrated no significant difference between EVs from uninfected and infected neurons (Fig 2B). Nanoparticle tracking analysis (NTA) was performed to determine the average size and concentration of EVs. Phosphate buffered saline (PBS) was used as a negative control [36,37]. EVs from C57BL/6 mouse embryo fibroblasts were used as a positive control at a known particle concentration (STEMON, 1x10^9 particles/mL). NTA allowed for visualization of the distribution of particle size in the control and experimental groups (Fig 2C). NTA quantification of EVs from uninfected and infected neurons demonstrated a significant decrease in particle concentration following infection (p < 0.005) (Fig 2D) with no significant difference in particle size (Fig 2D), consistent with the TEM data. Since NTA measures all particles and is not specific to EVs, a CD63 ELISA was conducted [38]. CD63 is one of three tetraspanin proteins, along with CD81 and CD9, commonly found on the surface of EVs and is widely used as an identifying EV marker [39,40]. CD63 quantification confirmed the presence of EVs from neuronal samples and that infection with *T. gondii* decreases CD63⁺ EV concentration (p < 0.05) (Fig 2E). To further confirm the presence of EVs in the two samples a Western blot probed for the presence of CD9, a tetraspanin protein, and Calnexin, an endoplasmic reticulum protein that should not be present in EV samples [41]. Western blots further confirmed the presence of EVs in both uninfected and infected neuronal samples (Fig 2F). Overall, these experiments determined that following Toxoplasma cyst infection of murine cortical neurons there is a significant decrease in neuronal production of EVs.

## 2.2. *T. gondii* infection alters the protein content of EVs

EVs contain multiple proteins, many of which are involved in the adhesion of the vesicle to the target cell [42] but many that can also reveal function and act as biomarkers of disease [43]. To determine if Toxoplasma infection alters the content of EVs, analysis of proteins via liquid chromatography (LC)-mass spectrometry (MS)/MS was conducted on uninfected and infected neuronal derived EVs. A principal component analysis (PCA) plot was generated to determine the grouping and relationship between EV samples from uninfected (blue) and infected (purple) neurons, demonstrating that protein signatures were different between the two samples (Fig 3A). When comparing protein similarities between the two groups there are 183 proteins that were consistently found in EVs from both uninfected and infected neurons. Only 17 proteins were uniquely identified in EVs from uninfected neurons, however, 101 proteins were found specifically in EVs from infected neurons (Fig 3B). Of all proteins only 20 were significantly differentially regulated and consisted of 6 proteins found only in EVs from infected neurons, and the remaining 14 found in both groups. Proteins significantly altered following infection were aligned to the mouse genome and a volcano plot was generated to visualize infection dependent protein changes (Fig 3C). There were 12 proteins significantly upregulated (red) in EVs from infected neurons, and 8 significantly downregulated (blue) (Fig 3C and S1 Table). Peptide verification was performed for three different protein sequences (S2 Fig). Highly upregulated proteins included vimentin (Vim), tubulin beta-4B chain (Tubb4b) and endoplasmin (Hsp90b1). These proteins have associations with the interferon response, innate immunity, stress response, and structural support (S1 Table). Tubb4b and Hsdp901 are commonly found in EVs, with Hsp90b1 being an identifying marker for isolated EVs [44,45]. Vim, a component of the cytoskeleton and vital for the regulation of intracellular signaling pathways [46] was expressed almost 8-fold higher in EVs from infected neurons. In contrast to the upregulation of immune associated pathways, the top 3 significantly downregulated proteins were involved with neuronal migration and growth (e.g., Reln, Chl1, Aplp1) (S1 Table). Reelin (Reln) is the most downregulated protein in EVs following infection and is linked to neuromodulation by protecting the brain against neurodegeneration and controlling synaptic plasticity [47]. Neuronal cell adhesion molecule L1-like protein (Chl1) and amyloid beta precursor like protein 1 (Aplp1) are involved in dendritic spine pruning and synaptic transmission [48,49].

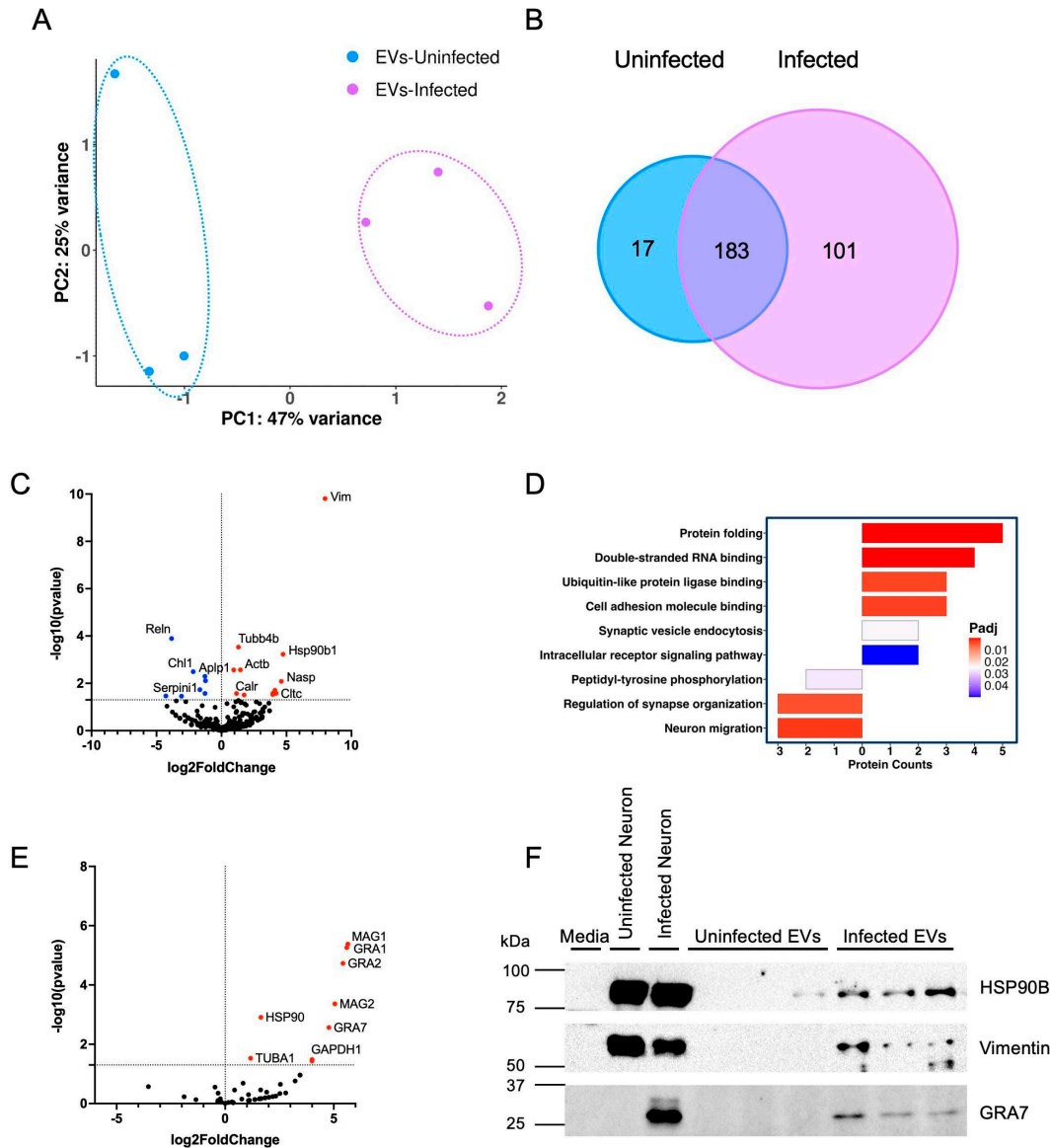

**Fig 3. Toxoplasma infection leads to changes in EV protein content. (A)** PCA (principal component analysis) of EVs from uninfected neurons (blue) and infected neurons (purple) after LC-MS/MS protein analysis. Three replicates per group. **(B)** Venn Diagram depicting the protein alterations following infection. **(C)** Volcano plot of protein changes aligned to the mouse genome. Above -log10(0.05) was considered significant. From the volcano plot 12 proteins are upregulated (red), 8 proteins are downregulated (blue), and 281 proteins are not changed (black). **(D)** GO analysis of Uninfected vs. Infected significant EV proteins. Bar graphs to the right of 0 indicate upregulated GO terms and to the left represent downregulated GO terms. Bar graphs are colored accorded to adjusted p value (padj). **(E)** Volcano plot comparing significant protein alterations following infection when comparing to the Toxoplasma genome. Anything above -log10(0.05) was considered significant. From the volcano plot 9 proteins are upregulated (red), 0 proteins are downregulated (blue), and 48 proteins are not changed (black). **(F)** Western Blot analysis of EV proteins from uninfected and infected neurons. Confirmation of LC-MS/MS protein data. Isolated EVs from infected neurons contain HSP90B (~83 kDa) exposed for 40 seconds, Vimentin (~53 kDa) exposed for 130 seconds, and GRA7 (~32 kDa) exposed for 104 seconds.

Gene ontology (GO) was performed on the significant proteins to help identify important functional properties related to protein changes. Thus, upon infection EV protein content suggests a downregulation of synapse organization and migration, and an upregulation in cell adhesion, binding, and intracellular signaling (Fig 3D).

Toxoplasma possesses sophisticated machinery to inject parasite proteins into host cells during and in the absence of host cell invasion, altering cell signaling events [50,51]. In addition, the parasite, itself a eukaryotic cell, is capable of EV production [22,23,52]. Thus, collection of EVs from infected neurons has the potential to contain parasite EVs and/or proteins excreted by the parasite and subsequently incorporated into host EVs. To identify any parasite specific proteins, alignment of protein sequences was also made to the Me49 strain of *T. gondii* (Fig 3E). This demonstrates the presence of 9 Toxoplasma proteins contained in neuronal EVs following infection. Some of these proteins are housekeeping proteins often incorporated into EVs (e.g., HSP90B, TUBA1, GAPDH1) [43], but others are *T. gondii* specific proteins known to be secreted effectors (e.g., MAG1, GRA1, GRA2, MAG2, and GRA7) involved in cyst wall formation and host cell manipulation [5,53,54] (Fig 3E). In addition, two cyst-wall specific proteins, MAG1 and MAG2, were present in EVs from infected neurons [53,55] (S2 Table). To validate these findings EV samples were probed by western blot for some of the top proteins identified by LC-MS/MS. Lysed uninfected and infected neurons were used as positive controls. The presence of a band for Vim, HSP90B, and Toxoplasma GRA7 was seen in EVs from infected neurons, consistent with the LC-MS/MS data (Fig 3F). These results demonstrate that infection of neurons with Toxoplasma alters EV content including the addition of parasite proteins.

## 2.3. Infection alters the miRNA content of EVs

The cargo of EVs can be diverse and includes DNA, RNA, and protein. Micro RNAs (miRNAs) can alter gene expression by post-transcriptional silencing or via chromatin modifications to silence or activate genes [56]. Previous work has demonstrated miRNAs (miR), specifically miR-124, from neuronal EVs regulate GLT-1 expression on astrocytes [18,57]. To determine if infection of neurons also alters the miRNA content of EVs, miRNA-Sequencing was conducted on EVs harvested from uninfected and infected neurons. miRNA transcript analysis determined distinct separation of EVs from uninfected (blue) and infected (purple) neurons, although with a broader variation of EVs from uninfected neurons. This indicates a parasite driven alteration in miRNA content (Fig 4A). Significantly expressed miRNA z-scores were normalized in order to obtain a heatmap to visualize the enriched miRNAs and observe similarities between the triplicates of each group and differences following infection (Fig 4B). When aligned to the mouse genome, 264 miRNAs were significantly differentially expressed with 8 miRNAs upregulated (red) and 6 miRNAs downregulated (blue) in EVs from infected neurons (Fig 4C). While some of the identified miRNAs do not have predicted gene targets and known functions, there are many that have been characterized. Of the downregulated miRNAs, both miR-3473b and miR-1224-5p have been reported to enhance inflammation and chemokine release in the brain [58,59] (S3 Table). In addition, miR-324-5p regulates excitatory synapse structure and function [60] and miR-466i-5p is predicted to induce neuronal apoptosis [61]. Of the upregulated miRNAs, miR-196a-5p, miR-199a-3p, miR-143-3p, miR-21a-5p have all been linked to an anti-inflammatory response through the inhibition of the NF-κB pathway [62–68] (S3 Table), with the exception of miR-29a-3p which exhibits a pro-inflammatory response through the phosphorylation of Akt and downstream NF-κB activation [69]. Of note, no significant difference was observed in miR-124 (miR-124-5p), responsible for GLT-1 regulation [17,18,70], suggesting other mechanisms are responsible for the reduction in astrocytic GLT-1 during infection. Thus, infection of neurons with Toxoplasma alters the small RNA content of EVs, specifically a downregulation of pro-inflammatory miRNAs, and an upregulation of anti-inflammatory miRNAs, however, does not support an infection-induced change in GLT-1 through miRNA-124 regulation.

## 2.4. EVs from infected neurons are taken up by astrocytes

Although cysts are primarily found in neurons in the brain, during Toxoplasma infection of an immunocompetent host, infection has the potential to have a much broader impact on neurological landscape via injection of parasite proteins into a cell that it does not invade [8] and as a result of the significant immune response necessary to control infection. Neurons have an intimate relationship with astrocytes and astrocytes can absorb neuronal-derived EVs, resulting in an alteration in their gene expression, protein levels, and immune function [17,18,70,71]. To determine if infection-induced EVs alter

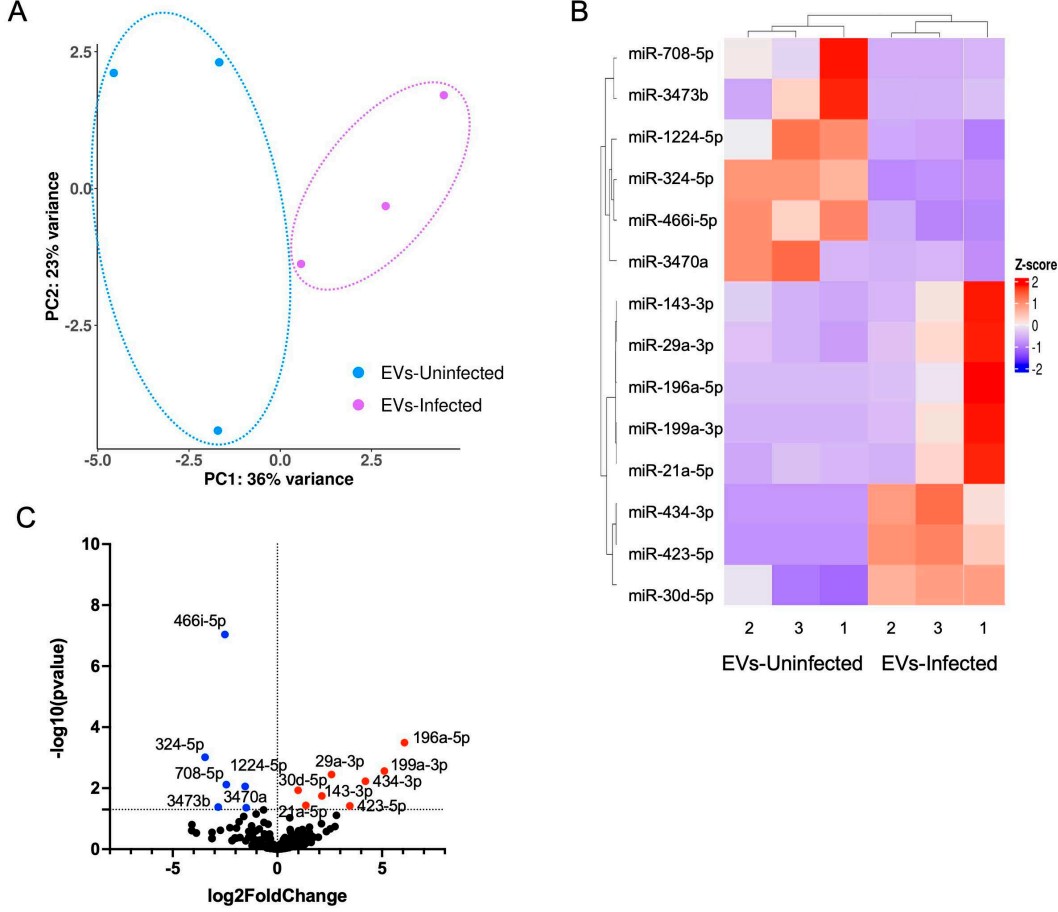

**Fig 4. Infection alters miRNA content of neuronal EVs. (A)** PCA of miRNA from isolated EVs of uninfected neurons (blue) and infected neurons (purple) after miRNA Sequencing. Three replicates per group. **(B)** Heat map of significant differentially enriched miRNAs after the isolation of EVs from Uninfected and Infected neurons. Each sample was conducted in triplicate. Each count was normalized, and the Z-score was mapped with columns and rows being clustered. **(C)** Volcano plot of differentially enriched miRNA comparing neuronal EV after infection. Anything above -log10(0.05) was considered significant. From the volcano plot 8 miRNA are upregulated (red), 6 miRNA are downregulated (blue), and 250 miRNA are not changed (black).

astrocyte gene expression, EVs from uninfected and infected neurons were incubated with primary murine astrocytes. Isolated EVs were stained with PKH67, a green lipophilic dye, to visualize uptake and location of EVs [72,73]. PKH67 labeled EVs colocalized in GFAP+ astrocyte cultures independent of the infection conditions of the neurons from which they were derived (Fig 5). EVs could be seen in the cytoplasm, often associated with vacuolar structures (Figs 5A and S3A). Interestingly, EVs could also be seen in the nucleus of astrocytes (Figs 5A and S3A). To further confirm the intracellular localization of EVs, immunocytochemistry for Toxoplasma derived GRA7 was conducted. GRA7 staining colocalized with PKH67 labelled EVs and was only present in cultures given EVs from infected neurons (Figs 5D and S3C). No EVs were visible in the absence of permeabilization supporting their intracellular localization. Of particular interest, GRA7 was also noted in the nuclei of cells containing GRA7+EVs and not in neighboring cells that had no EVs (Fig 5D). Our data suggests a significant alteration in protein content of EVs from infected neurons which may influence EV uptake by target cells. Quantification of EV localization was conducted to determine if the infection status of the neurons altered EV uptake and cellular localization within astrocytes. Infection did not lead to a difference in the proportion of EVs taken up by astrocytes (Fig 5B), nor did it alter their nuclear or cytoplasmic localization (Fig 5C).

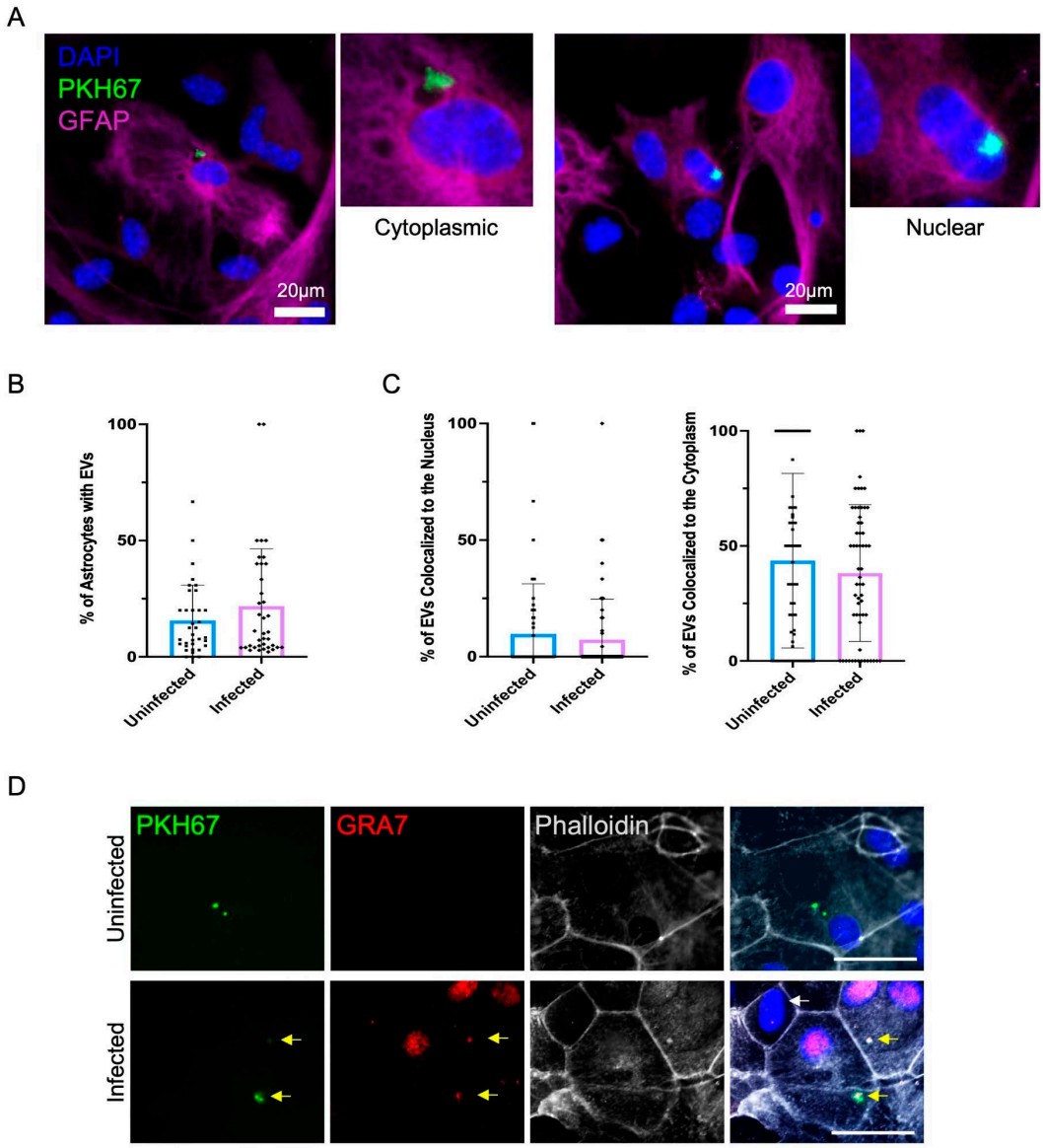

**Fig 5. Astrocytes take up neuronal EVs.** For quantifications "n" represents a single cell throughout multiple experiments. **(A)** Fluorescent microscopy of astrocytes after the addition of EVs from uninfected and *T. gondii* infected neurons indicating the location of EVs in the cytoplasm versus nucleus. DAPI stains the nucleus, PKH67 stains the EVs, GFAP stains the astrocytes. Scale bar: 20 μm. **(B)** Quantification of the percentage of astrocytes that contain EVs (Unpaired t-test, n (Uninfected) = 35, n (Infected) = 38, Uninfected vs. Infected p = 0.2081). **(C)** Quantification of the percentage of EVs that are colocalized with the nucleus of the astrocytes (Unpaired t-test, n (Uninfected) = 61, n (Infected) = 59, Uninfected vs. Infected p = 0.4900). Quantification of the percentage of EVs that are colocalized with the cytoplasm of the astrocytes (Unpaired t-test, n (Uninfected) = 61, n (Infected) = 59, Uninfected vs. Infected p = 0.3853). **(D)** Permeabilization experiments on EVs and GRA7 protein. Yellow arrows are points to PKH67 + GRA7 + parasites that contain GRA7 + astrocyte nuclei. The white arrow indicated the lack of GRA7 + EVs and therefore lack of GRA7 + astrocyte nuclei. PKH67 stains the EVs, GRA7 stains the internalized GRA7 proteins, Phalloidin stains the actin filaments of the astrocyte. Scale bar: 30 μm.

Having demonstrated astrocytes can uptake EVs from both uninfected and infected neurons we were interested in the EV uptake mechanism something that is poorly understood in astrocytes. Previous studies have indicated that microglia take up oligodendrocyte derived EVs through the macropinocytosis pathway [74,75]. To better understand the mechanism of neuronal EV uptake by astrocytes, endocytic pathways were blocked and EV uptake measured. Two different endocytic

pathway inhibitors were added to astrocyte cultures, 1 hour prior to the addition of EVs (S3B Fig). When blocking macropinocytosis by both EIPA and cytochalasin D, EV uptake was still observed in astrocytes from both EV groups. This data suggests that neuronal derived EVs are not taken up by astrocytes through macropinocytosis and alternative mechanisms such as clathrin mediated or caveolin endocytosis need to be considered (S3B Fig). Altogether, these data demonstrate that EVs from uninfected and infected neurons are internalized by astrocytes via a macropinocytosis-independent pathway and localize to both the cytoplasm and astrocytic nucleus.

## 2.5. EVs alter astrocyte gene expression and GLT-1 protein expression in an infection-dependent manner

Our previous study demonstrated a reduction in astrocytic GLT-1 in the brain of infected mice and that this reduction was, at least in part, responsible for neuronal pathology. GLT-1 loss was not directly correlated to areas of infected astrocytes [10]. Previous work supports the concept that neurons communicate and can regulate astrocyte function via EV release, including the expression of GLT-1 [17,18,70]. In addition, our observation of GRA7 in the nuclei of EV containing astrocytes suggests the ability of parasite proteins to enter this site and potentially alter cell responses. To test this and determine if infection of neurons can alter this avenue of communication, EVs from uninfected and infected neurons (2 µg/mL) were incubated with astrocytes and astrocyte gene expression measured via RNASeq. Astrocytes with the addition of media alone were used as a baseline control. *In vitro* derived cultured astrocytes do not express GLT-1 therefore Dexamethasone (Dex) was added to increase GLT-1 expression, acted as a positive control for GLT-1 and allowed us to determine the ability of EVs from infected neurons to downregulate GLT-1 [76,77]. In addition, astrocytes stimulated with LPS and IFNγ, known to induce a significant pro-inflammatory response in the astrocyte, acted as a positive control for known changes in gene expression [78]. Analysis of overall gene expression in replicates and stimulations via PCA verified the concise groupings of EV samples in our negative and positive controls (Fig 6A). Although, replicates grouped together, changes between uninfected and infected EVs were dwarfed by the dramatic changes between the positive and negative controls. When zooming in and focusing only on the effects of the addition of EVs from uninfected and infected neurons, the PCA confirmed replicates clustered together and were distinct from each other (Fig 6B). These small, but significant differences between EV stimulations were retained even if the groups were re-analyzed excluding the controls (S4A and S4B Fig).

A comparison of DESeq2 normalized counts between the five groups confirmed an increase in GLT-1 gene expression (*Slc1a2*), with the addition of Dex (Fig 6D). However, there is no difference in GLT-1 gene expression between the addition of uninfected versus infected EVs (Fig 6D). In addition to inhibition of transcription, GLT-1 can also be regulated by post-translational modifications [19]. To determine if EVs from infected neurons alter GLT-1 protein expression a western blot for GLT-1 was conducted using the same conditions as before (Fig 6E and 6F). Quantification of the normalized bands (GLT-1/β-Actin) identified a significant decrease in GLT-1 protein intensity after the addition of EVs from infected neurons (p < 0.005) (Fig 6F). Further experiments indicated the decrease in GLT-1 protein expression was not dependent on the concentration of EVs added or the MOI of the parasite, leading us to hypothesize that there may be post-translational modifications at play and the decrease is dependent on EV content (S5 Fig).

To expand our understanding of the effect of neuronal infection on astrocyte gene transcription, significantly expressed DEG z-scores were normalized and visualized on a heatmap to determine genes that are highly enriched in astrocytes after the addition of EVs from uninfected versus infected neurons (Fig 6C). This revealed a similarity within biological replicates and striking patterns of genes that were significantly different. Significance was determined as a p-value greater than the negative log of 0.05 and a log2FC (fold change) greater or less than 0. Analysis of differentially expressed genes (DEGs) with a p-value<0.05 were plotted to identify those significantly changed in response to EVs from infected neurons (Fig 6G). While a majority of the genes were not altered (31,135 genes), there were 227 genes upregulated, and 176 genes downregulated following addition of EVs from infected neurons. The most significantly upregulated genes (e.g., *Ifit1*, *Usp18*, *Rtp4*) are all involved in a pro-inflammatory innate immune response. Interferon-induced protein with tetratricopeptide repeats 1 (*Ifit1*) induces type I interferon signaling [79], while the ubiquitin specific peptidase 18 (*Usp18*)

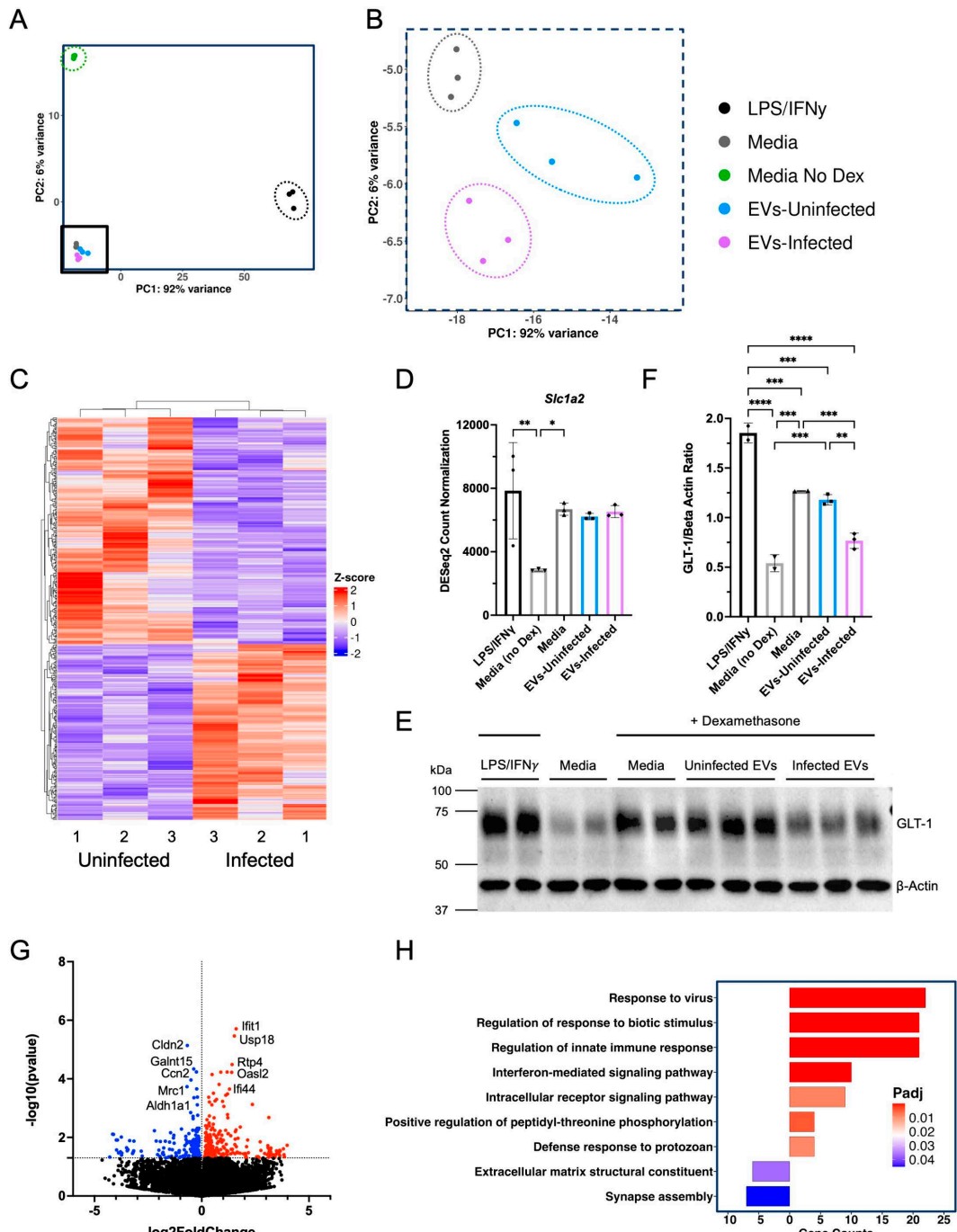

**Fig 6. EVs from infected neurons alter astrocyte gene expression and decrease GLT-1 protein expression. (A)** PCA of astrocytes after 24 hours with the addition of LPS/IFNγ (black), cell culture media (gray), cell culture media without dexamethasone (green), EVs from uninfected neurons (blue) and EVs from infected neurons (purple). **(B)** Zoomed in PCA plot focusing on Uninfected EVs and Infected EVs. **(C)** Heat map of significant astrocyte DEGs after the addition of EVs from Uninfected EV and Infected EVs. Each sample was conducted in triplicate. Each count was normalized, and the Z-score was mapped with columns and rows being clustered. **(D)** *Slc1a2* gene expression levels of astrocytes after the addition of LPS/IFNγ, astrocyte media without dexamethasone, astrocyte media, EVs from uninfected neurons, and EVs from infected neurons. Comparison of *Slc1a2* expression (gene for GLT-1), using DESeq2 Normalized counts (One-way ANOVA, n = 3, LPS/IFNγ vs. Media (no Dex) p value = 0.0086, LPS/IFNγ vs. Media p value = 0.8350, LPS/IFNγ vs. EVs-Uninfected p value = 0.6140, LPS/IFNγ vs. EVs-Infected p value = 0.7683, Media (no Dex) vs. Media p value = 0.0418, Media (no Dex) vs. EVs-Uninfected p value = 0.0795, Media (no Dex) vs. EVs-Infected p value = 0.0516, Media vs. EVs-Uninfected p value = 0.9928,

Media vs. EVs-Infected p value = >0.9999, EVs-Uninfected vs. EVs-Infected p value = 0.9984). **(E)** Western blot for GLT-1 (~62 kDA) protein expression. β-Actin (~42 kDa) used as loading control. Image of gel was taken 300 seconds after exposure. **(F)** Quantification of Western Blot bands using Fiji (ImageJ). A ratio of the pixels (GLT-1/β-Actin) was taken and plotted (One-way ANOVA, n (LPS/IFNγ) = 2, n (Media without Dex.) = 2, n (Media) = 2, n (EVs-Uninfected) = 3, n (EVs-Infected) = 3, Media without Dex. vs. Media p value = 0.0001, Media without Dex. vs. EVs-Uninfected p value = 0.0002, Media without Dex. vs. EVs-Infected = 0.0325, Media vs. EVs-Uninfected p value = 0.4825, Media vs. EVs-Infected p value = 0.0006, EVs-Uninfected vs. EVs-Infected p value = 0.0010). (LPS/IFNγ vs. Media (no Dex) p value = <0.0001, LPS/IFNγ vs. Media p value = 0.0005, LPS/IFNγ vs. EVs-Uninfected p value = 0.0001, LPS/IFNγ vs. EVs-Infected p value = <0.0001, Media (no Dex) vs. Media p value = 0.0001, Media (no Dex) vs. EVs-Uninfected p value = 0.0002, Media (no Dex) vs. EVs-Infected p value = 0.0545, Media vs. EVs-Uninfected p value = 0.6531, Media vs. EVs-Infected p value = 0.0007, EVs-Uninfected vs. EVs-Infected p value = 0.0012). **(G)** Volcano plot of DEGs comparing astrocytes RNA after the addition of EVs from uninfected neurons vs. the addition of EVs from infected neurons. Anything above -log10(0.05) was considered significant. From the volcano plot 227 genes are upregulated (red), 176 genes are downregulated (blue), and 31,135 genes are not changed (black). **(H)** GO analysis of Uninfected EV vs. Infected EV DEGs. Bar graphs to the right of 0 indicate upregulated GO terms and to the left represent downregulated GO terms. Bar graphs are colored accorded to adjusted p value (padj).

and receptor transporter protein 4 (*Rtp4*) negatively regulates type I interferon signaling [80–83]. The top significantly downregulated genes are associated with cellular communication (e.g., *Cldn2*, *Galnt15*, *Ccn2*) (S4 Table). These genes act in cellular binding and adhesion and all are important for allowing the attachment and entry of molecules, with claudin 2 (*Cldn2*) allowing for paracellular permeability [84]; polypeptide N-acetylgalactosaminyltransferase 15 (*Galnt15*) predicted to enable carbohydrate binding [85]; and cellular communication network factor 2 (*Ccn2*) permits cell adhesion [86], indicating that the addition of EVs from infected neurons is reducing the adhesive properties of astrocytes. GO analysis confirmed the upregulation of immune responsiveness of astrocytes including the upregulation of 20 genes involved with "regulation of innate immune response" and "response to a virus" (Fig 6H). In addition, the GO terms "defense response to protozoan" and "interferon-mediated signaling pathway" were identified in cultures that received EVs from infected neurons compared to EVs from uninfected neurons. This signature differed in magnitude and function to stimulation with LPS and IFNγ (S4C Fig) and signifies the specificity achieved by EVs from infected neurons, despite no actual parasite being in these cultures (Fig 6H). Overall, this data reveals the potential for infected neurons to regulate the inflammatory actions of astrocytes via EVs and to communicate the presence of intracellular Toxoplasma.

Our lab has recently characterized astrocyte subpopulations *in vivo* over the course of Toxoplasma infection [87]. To determine if the astrocytic transcriptional changes seen *in vitro* also occur *in vivo*, we compared upregulated genes from our infected EV samples to differentially regulated genes from astrocytes following *in vivo* infection (Fig 7). By combining astrocytes from naïve, acute and chronically infected time points, 8 clusters of astrocytes were observed with Clusters 2 and 5 found predominantly in naïve mice (Fig 7A). Gene expression within each cluster compared to all astrocytes was visualized in a dot plot (Fig 7B). Almost all genes upregulated by EVs from infected neurons are only found following infection, confirming our *in vitro* astrocyte phenotypes also occur with Toxoplasma infection *in vivo*.

Addition of EVs from infected neurons led to pro-inflammatory changes in astrocytes (Fig 6H). To determine if there are populations of astrocytes with a similar profile, we identified which clusters our top expressing genes were found in during the chronic 42 days post infection time point (Fig 7A–7C). Most EV-regulated genes were found in Clusters 1 and 4. Upregulated genes were also found in Cluster 7 and 8, however these clusters were comprised of very few cells and were not analyzed further. Genes strongly upregulated *in vitro* such as *Rtp4* and *Ifit1* were not confined to one cluster *in vivo* but rather found in all clusters except the naïve associated Cluster 5 (Fig 7C). Other genes appeared to be confined mainly to Clusters 1 and 4, such as antiviral and immunity genes *Gpb5* and *Stat2* [88,89], and interferon stimulating genes involved with restricting virus replication, *Ifi44* and *Irf7* [90,91]. A set of upregulated *in vitro* and *in vivo* genes following infection were also strongly confined to Cluster 1 including *Usp18*, *Tmem17* involved with tumor progression, and antiviral gene *Ddx60* [92,93] (Fig 7C). Interestingly, interferon induced gene *Cxcl9* was strongly expressed in Cluster 4 [94]. Lastly, gene ontology analysis of Clusters 1 and 4 revealed multiple terms that overlapped with the signature generated by *in vitro* stimulation of astrocytes with EVs from infected neurons, including "response to virus", "regulation of innate immune

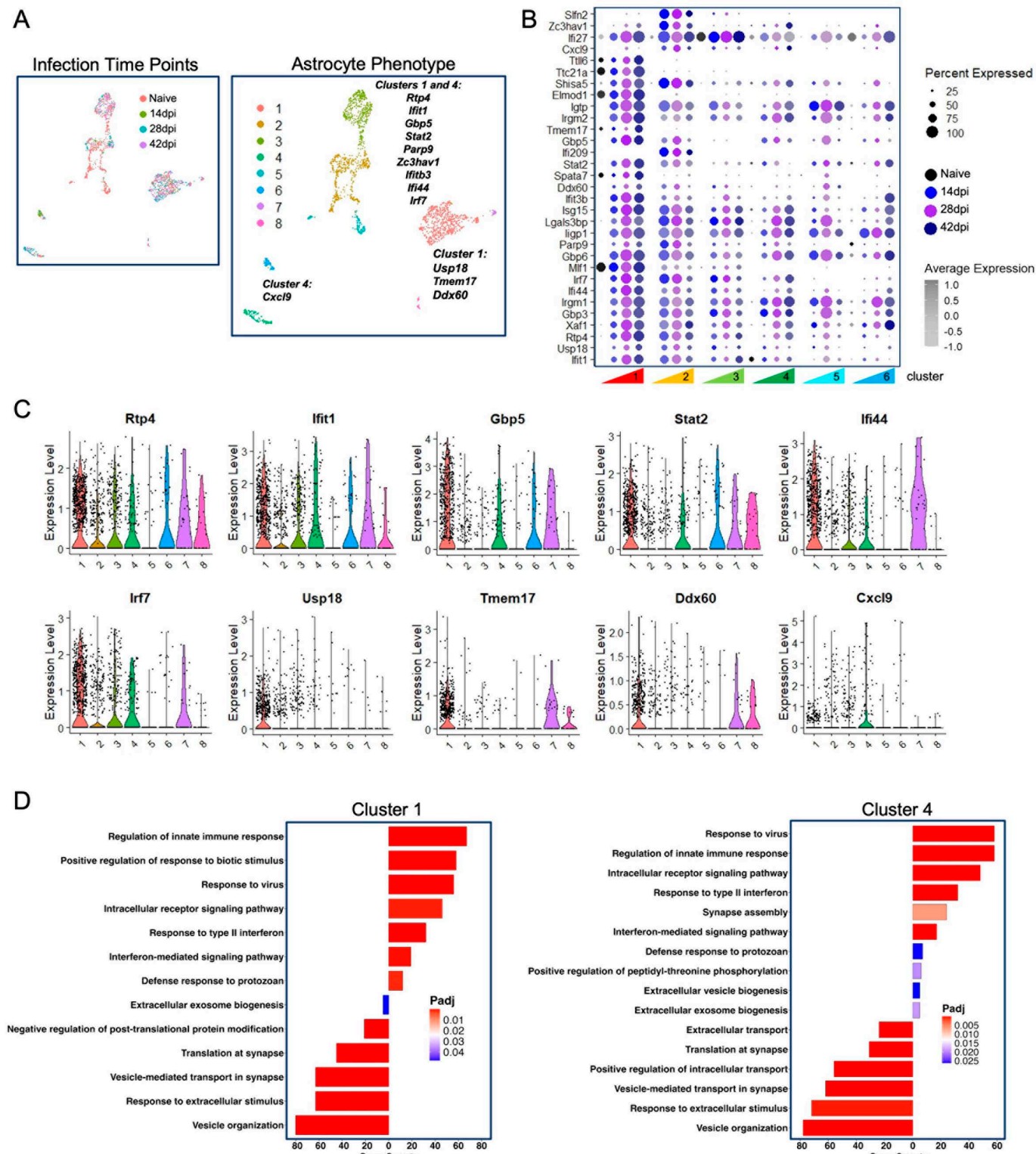

**Fig 7. Astrocytes exposed to EVs from infected neurons *in vitro* experience similar transcriptional changes to astrocytes following infection *in vivo*. (A)** UMAP plot of astrocytes identifies 8 transcriptionally defined subpopulations (Clusters 1-8) with lists of upregulated *in vitro* genes listed that are found in the data. **(B)** Dot plot displaying the average expression levels of marker genes identified in each cluster. Each dot color corresponds to a

time point: naïve (black), acute 14 dpi (blue), early chronic 28 dpi (purple), and chronic 42 dpi (dark blue). The size of each dot indicates the percentage of cells within a cluster that express each gene, while the color intensity represents the average expression level of each gene across all cells within a cluster, with the darkest shade indicating the highest expression level. **(C)** Violin plots of strongly expressed genes in Clusters 1 and 4 (*Rtp4, Ifit1, Gpb5, Stat2, Ifi44, Irf7, Usp18, Tmem17, Ddx60, Cxcl9*). **(D)** GO analysis showing enriched biological pathways and the associated cell counts for Cluster 1 and Cluster 4. Bar graphs to the right of 0 indicate upregulated GO terms and to the left represent downregulated GO terms. Bar graphs are colored accorded to adjusted p value (padj).

response", and "defense response to protozoan" (Fig 7D). Of note upregulated GO terms also include "extracellular vesicle biogenesis" and "extracellular exosome biogenesis."

## 3. Discussion

Here we demonstrate that extracellular vesicles (EVs) can be harvested from Toxoplasma cyst-containing cortical neuron cultures and that infection of neurons leads to a decrease in EV production and a fundamental change in their content. Significantly, 9 parasite proteins, including dense granule proteins known to be secreted effectors, are detected in EVs harvested from cyst-containing neurons. Intact labelled EVs were taken up by astrocytes and observed in the cytoplasm and nucleus of the host cell with parasite-derived GRA7 found within intracellular EVs and in the host cell nucleus. Furthermore, incubation of astrocytes with EVs from infected neurons led to a significant change in astrocyte transcriptional signatures, stimulating a pro-inflammatory immune response and downregulating the glutamate transporter GLT-1.

This study reveals the presence of Toxoplasma cysts leads to a decrease in either EV production or release by neurons. All eukaryotic cells release EVs [95,96], therefore infection of neurons could have led to an increase in the concentration of EVs with multiple cells – parasites and neurons – contributing to their production [97–100]. A potential reason for a decrease in EV's harvested from infected neurons is that infection is causing a change in neuronal health and function. Indeed, this occurs *in vivo* as reported by several groups [8,10,101,102]. These short-term *in vitro* cultures, long enough for cyst formation, do not lead to the death of neurons and markers of neuronal viability are intact, however a reduction in dendrite formation can be seen by immunocytochemistry (Figs 1A and S1). In addition, our data revealed the downregulation of miR-324-5p in EVs following infection (Fig 4 and S3 Table), which when knocked out in mice can lead to a decrease in dendritic spine density and morphology in the brain [60]. Thus, infected neurons are retracting dendrites, becoming less morphologically complex and such a process, involving a reduction in surface area and membrane modifications may also be associated with reduced EV release.

Our finding of reduced EV production from infected cells is also supported by studies from the Carruthers group demonstrating Toxoplasma's ability to sequester host ESCRT proteins in a GRA14 dependent manner [103,104]. The ESCRT pathway, involved in autophagy and cell division, is also required for extracellular vesicle formation and release [105]. Thus, our finding of reduced EV production from cyst-containing neurons would support the concept that the parasite is reducing the ability of the host cell to operate this machinery by diverting ESCRT proteins for its own use. Although these prior studies showed the highest enrichment of GRA14 bound host proteins in HFF cells, neuronal infection with tachyzoites also led to components of ESCRT I and III binding. Our data would support the ability of Toxoplasma to reduce ESCRT host activity in neurons, even as a slow growing cyst. Alternatively, infection could lead to a reduction in EVs from neurons independent of direct parasite manipulation. Indeed, only half of the neurons in these cultures are infected. Production of EVs can also lead to autocrine uptake [95]. Therefore, an increase in EV uptake by the infected neuron, would also lead to this overall decrease in EV concentration. The mechanism of EV uptake by neurons is unknown but our analysis of protein content of the EVs suggests an increase in "synaptic vesicle endocytosis," from the upregulation of actin, beta (Actb) and clathrin heavy chain (Cltc) (Fig 3D), and many proteins involved in the uptake of synaptic vesicles are also involved in the numerous potential mechanisms of EV uptake [106].

There is an assumption that the measurement of proteins and small molecules in EVs directly and proportionally reflects the content and function of the cell with proteins being indiscriminately packaged into the vesicle. However, there is evidence of selectivity and sorting of cargo prior to EV release, for example the specific exclusion of pro-inflammatory mitochondrial proteins [107,108]. In addition, the purported function of EVs as an important mechanism of cell-cell communication suggests a need for packaging of cargo to be targeted and selective. Our investigation of the content of EVs from infected neurons revealed significantly altered protein and miRNA content. Heat-shock proteins are common stress response regulators and are also protein markers of EVs. Our data reveals their presence and an increase in HSP90 and Hsp90b1 following infection [109–111]. In addition, we are further able to confirm that these are neuronal EVs through the presence of neural cell adhesion molecule L1-like protein (Chl1) [112], the presence of CD9, and the lack of the ER marker Calnexin (Fig 2F). While many of the proteins were not modified upon infections, there were several proteins that were significantly up or downregulated. Overall the changes in EV protein content following Toxoplasma infection suggests a downregulation of neuronal organization and migration, picked up by two downregulated GO terms and includes the 4-fold reduction in reelin (Reln), a glycoprotein particularly important for neuronal migration during development [113,114]. Morphological alterations and a decrease in neuronal migration is consistent with previously published work [115] and is supported by the appearance of neurons in culture (Fig 1) and the contraction and decrease in neuronal complexity *in vivo* [8,10].

Infection led to EV content that suggests the suppression of inflammation. MicroRNAs are frequently inhibitory but can also be activators of gene transcription [116]. The miRNA content of EVs from infected neurons is overwhelmingly associated with blocking NF-κB and other pro-inflammatory pathways. Both the downregulation of miR-3473b [59], and upregulation of miR-199a-3p, miR-143-3p and miR-21a-5p [63,65–68] leads to a reduction in inflammatory signals. Notably, miR-199a-3p, upregulated over 5-fold in EVs from infected neurons, can suppress astrocyte activation and NF-κB-mediated inflammation [64], while miR-143-3p negatively regulates NOTCH signaling and in this context may, as a result, decrease cell proliferation, differentiation and BBB integrity [117,118]. miRNA-143-3p and miRNA-21a-5p are known to be modulated during Toxoplasma infection in macrophages and microglia EVs, and are predicted to downregulate nitric oxide (NO) synthase [119–121]. A host protein only identified in EVs from infected neurons and highly upregulated is vimentin (Vim), an intermediate filament protein found in the cytoskeleton that inhibits type I interferon production [46], and in mouse models of spinal cord injury is a key regulator in reactive astrocytes [122].

By using the resources available on ToxoDB (https://toxodb.org) we were also able to determine one of the most revealing aspects of EV content, namely the presence of parasite specific proteins. A total of 8 parasite proteins were identified in EVs from infected neurons. Three of these, HSP90, TUBA1 and GAPDH1, are found in EV's from many eukaryotic cells. However, an additional 5 proteins, GRA1, 2, 7 and MAG1 and 2 play a role in cyst formation but importantly are secreted by the parasite and many have the potential to manipulate the host immune response. GRA proteins help form the intravacuolar network (IVN), cyst wall and the inner cyst matrix [123]. Our EVs are harvested after 3 days of cyst formation and therefore these cysts are relatively immature. MAG1, the most abundant parasite derived protein in our EVs is secreted and can suppress inflammasome activation [124]. MAG1 and MAG2, both components of the cyst matrix, are seen by the immune system as evidenced by the significant anti-MAG immune response [54,125,126]. Importantly, similar proteins seen during cyst development (e.g., GRA12) were not found in our EVs suggesting either that these do not escape the cyst or that there is selectivity in the parasite-derived cargo of EVs. GRA1, found at one of the highest levels in EVs, at this early time point of cyst development is found on the outer layer of the cyst, and is not pulled down in the presence of CST1 (carbohydrate components of the tissue cyst) and is dispensable for cyst growth and stability [123,127]. This and the presence of GRA1 in EVs could support GRA1 being a protein that is secreted by bradyzoites and interacts with the host cell rather than playing a critical structural or developmental role. GRA2 is found preferentially in the cyst wall matrix and remains there throughout the maturation of the cyst [123]. Both GRA1 and GRA2 are associated with increased cyst survival in the presence of T cells [128] supporting a role for host cell manipulation and suppression

of immunity. Indeed, GRA7, the third most abundant GRA in EVs from infected neurons, is a protein known to interact with the host, and GRA1 [126,129]. GRA7 is also one of the dense granule proteins, alongside GRA1, 3, and 14, found in infected cultures in "bead" formation or BOAS (beads-on-a-string); membranous extensions of the parasitophorous vacuole (PV) that can connect to multiple PVs and to host nuclei [129–131]. This appearance is consistent with an EV function albeit with a potentially unique parasite secretory process. The presence of parasite proteins in EVs collected from infected neurons could be due to two mutually independent models. Firstly, parasite proteins are secreted across the PV and incorporated into host cell EVs, or secondly, the parasite produces EVs that are either released independently or alongside host EVs. Interestingly, probing EVs for GRA7 in astrocyte cultures revealed not all EVs from infected neurons contained GRA7 (S3C Fig). This would support the presence of the second model of EV release with EVs from infected neurons composed of parasite and host cell EV populations. Suggesting, at least a portion of, GRA7 is incorporated in parasite EVs that cross four membrane structures: the parasite membrane; the PV; the donor cell membrane; and the recipient host cell membrane.

Our data show that labelled EVs are taken up by astrocytes in culture and colocalize not only to the cytoplasm but also the nucleus of the cell (Figs 5 and S3). Other studies have reported observing EVs within the nucleus with incorporation into late endosomes being the likely mechanism [132,133]. Although we have ruled out that initial uptake of neuronal EVs by astrocytes is via macropinocytosis (S3B Fig) [74,75], further studies will need to be conducted to determine the uptake mechanism. Our observation that GRA7 is found specifically in the nuclei of cells containing GRA7$^+$ EVs and not in neighboring cells (Fig 5C) strongly suggests transport of parasite-proteins to host cells via EVs leads to direct changes in host cell transcription. There is no evidence of direct DNA binding by GRA7, although it is phosphorylated by the host cell [129].

EVs from infected neurons generate a specific transcriptional signature in astrocytes. Analysis of changes in transcription induced by EVs was originally to determine whether infection of neurons could indirectly lead to the downregulation of GLT-1 in astrocytes via changes in neuronal EVs [10,17,18]. Although there was no change in miRNA-124 in the EVs following infection, indicating that GLT-1 regulation is at least partially miRNA-124 independent, we did demonstrate that the addition of EVs from infected neurons significantly decreases GLT-1 protein expression in primary astrocytes. This, combined with the overall reduction in EV production supports the origin of GLT-1 downregulation being infected neurons.

Despite the content of EVs suggesting their potential to limit inflammation, addition of EVs from infected neurons altered astrocyte gene expression towards a pro-inflammatory/anti-parasitic immune response including the upregulation of interferon gamma induced GTPase (*Igtp*) and interferon inducible GTPase 1 (*Iigp1*). These changes differed dramatically in magnitude to LPS and IFNγ stimulation. Most importantly the signature of these transcriptional changes is quite different. The top GO terms in response to LPS and IFNγ are focused on generation of RNA and protein ("ribonucleoprotein biogenesis") followed by "regulation of innate immune response" and "response to virus." In comparison the dominant transcriptional signature in astrocytes incubated with EVs from infected neurons is about chemotaxis ("chemotaxis"; "leukocyte migration"). The "defense response to protozoan," a particularly striking finding when comparing EVs from infected versus uninfected neurons, is also an upregulated GO term in response to LPS/IFNγ. This should be expected with IFNγ a required signal for any immune response against protozoa. It is the specificity achieved by EVs from infected neuron cultures where there is no IFNγ present that makes our EV findings important. Astrocytes are able to control parasite replication through the activation of STAT1 (signal transducer and activator of transcription 1) and the use of IRGs (immunity regulated GTPases) [134,135] and are thus very adept at responding to Toxoplasma. It implies that the EV parasite-derived cargo of GRA proteins, instead of manipulating the host cell are being detected in a classical immune mediated mechanism either by pattern recognition receptors or inflammasome activation. Alternatively, the combined changes in host neuronal proteins and nucleic acids in the EVs could be indicating that the source cell requires an astrocytic change in function from a supportive neurotransmitter regulating cell to a damage control agent. The identification of two populations of astrocytes *in vivo* that mirror the phenotype of infected EV stimulated *in vitro* astrocytes helps validate this

transcriptional signature as infection-associated and functionally relevant. Clearly the direct comparison is an oversimplification of what must be occurring in the brain where astrocytes receive signals from multiple cytokines, chemokines, and cell-cell interactions that have the potential to generate a far more amplified response (as seen in LPS/IFNγ stimulation *in vitro*). However, the similarity between these *in vitro* and *in vivo* signatures supports a potential role for EVs from cyst-infected neurons to trigger immune responses during early infection in the brain, maintain inflammatory signatures during chronic infection or stimulate an anti-cyst response mediated by peripheral immune cells [136,137]. This study tested the effect of neuronal EVs on astrocytes due to the known close relationship between these cells and the specific questions on the infection-induced changes in GLT-1. However, EVs derived from cyst containing neurons could also be taken up by many other cells in the brain including microglia and peripheral immune cells and therefore this alteration in host EV production and content in a few infected neurons could have a much broader impact on neuroinflammation in the brain.

Lastly, the use of EVs as known biomarkers for disease, especially as readouts of what is taking place in the brain, is a growing possibility [138]. EVs generated in the CNS can be detected in cerebral spinal fluid, blood, tears, and urine. As we have seen in our own data, they provide cell specificity with neuronal derived EVs containing Chl1. Toxoplasma infection is determined by the presence of peripheral circulating antibodies that would be generated no matter the location of infection. A large body of work has attempted to correlate seropositivity with other unrelated neurological conditions despite not knowing whether infection includes cysts in the brain [139]. The demonstration that EVs from infected neurons contain stage-specific parasite proteins provides a potential screen for the presence of neuronal cysts and a refinement of the broader effects of Toxoplasma infection on brain neurochemistry and clinical disease.

Overall, our data suggest that the presence of Toxoplasma cysts within neurons has the potential to broadly change the brain environment through an alteration in the production and content of extracellular vesicles. The dispersion and uptake of these EVs drives an infection-induced astrocyte phenotype promoting inflammatory signaling at the expense of neurotransmitter uptake.

## 4. Materials and methods

### 4.1. Ethics statement

All mice were stored in accordance with the Animal Welfare Act and all efforts were made to minimize suffering. All protocols were approved by the Institutional Animal Care and Use Committee (IACUC) at the University of California, Riverside. SW, CBA, and C57BL/6 mice were obtained from Jackson Laboratories (Jackson ImmunoResearch Laboratories, Inc., West Grove, PA, USA) and maintained in a pathogen-free environment according to IACUC protocols at the University of California Riverside.

### 4.2. Cell culture

Primary cortical neuron cultures were grown from the cortex of C57BL/6 embryos (E18-20). The forebrain was removed, and meninges stripped in Hanks' Balanced Salt Solution (HBSS) without $Ca^{2+}$ and $Mg^{2+}$. The tissue was spun at 300 g for 2 minutes at room temperature. Neural tissue dissociation kit (Miltenyi Biotec, 130-093-231) was used according to the manufacturer's protocol. Tissue was incubated with enzymes at 37°C with 5% $CO_2$ under slow, continuous rocking conditions and dissociated through an 18-gauge needle followed by a 20-gauge needle followed by further incubation. The tissue was filtered through a 40 μm strainer and washed in HBSS with $Ca^{2+}$ and $Mg^{2+}$. Cells were spun at 300 g for 10 min at room temperature and the pellet was resuspended in neurobasal medium (supplemented with 2% B-27 supplement, 0.5 mM L-glutamine, and 1% penicillin-streptomycin). Cells were plated on poly-L-lysine (PLL) coated 6 well plates at $5.0x10^5$ cells per well in 2 mL and incubated at 37°C with 5% $CO_2$ in complete neurobasal medium supplemented with 25 M of L-Glutamic Acid. Half of the medium was changed every three days with neurobasal medium and filtered through a 0.8 μm filter and stored at -80°C. DOI: https://doi.org/10.17504/protocols.io.3byl4wpyovo5/v1.

Primary cortical astrocyte cultures were grown from the cortex of C57BL/6 pups (P0-3). The forebrain was removed, and meninges stripped in an isolating medium (1X PBS with 0.1% BSA (bovine serum albumin) and 0.45% glucose). Cortices were mashed and filtered through a 40 μm strainer in washing medium (Dulbecco Minimum Essential Medium (DMEM) supplemented with 2% HI-FBS (heat-inactivated fetal bovine serum)). The cells were spun at 2,000 rpm for 10 minutes at 4°C and washed three times. The final pellet was resuspended in astrocyte culture medium (DMEM supplemented with 10% HI-FBS, 1% non-essential amino acids, 1% L-glutamine, 1% 4-(2-hydroxyethyl)-1-piperazineethanesulfonic acid buffer (HEPES), and 1% penicillin/streptomycin) and plated in T25 flasks incubated at 37°C with 5% $CO_2$. Astrocyte medium was changed every 2 days until day 8. On day 8 the flasks were shaken for 2 hours at 260 rpm at 37°C. The astrocyte medium was changed, and the flasks were then shaken for 24 hours at 100 rpm at 37°C. The cells were washed with HBSS without Ca$^{2+}$ and Mg$^{2+}$ and then lifted with 0.25% Trypsin-EDTA (disodium ethylenediaminetetraacetic acid). The reaction was stopped with astrocyte medium, and the cells were centrifuged at 1,200 rpm for 5 minutes at room temperature. The pellet was resuspended in astrocyte medium and plated on 6 well plates at 1.0x10$^5$ or 2.5x10$^5$ cells/mL in 2 mL and incubated at 37°C with 5% $CO_2$. DOI: https://doi.org/10.17504/protocols.io.q26g7moj3gwz/v1.

## 4.3. Infections

The Me49B7 strain of *T. gondii* was maintained in cyst form through continuous passage in SW and CBA background mice. For *in vitro* cultures the parasite was maintained in human foreskin fibroblasts (HFF) in parasite medium (DMEM supplemented with 5% HI-FBS, 1% L-glutamine, 1 mL/L gentamicin, and 1% penicillin/streptomycin) and incubated at 37°C with 5% $CO_2$. The seeding of embryonic cortical stem cells does not lead to 100% conversion to neurons. Indeed from 5 million plated cells ~10, 000 fully develop to cortical neurons. These counts were made by direct microscopic examination and did not require harvesting of cells thereby avoiding cell loss. Cell number remained the same for the length of the culture period and following infection. Once mature the neurons were infected with the parasite at day 8. Different MOIs (multiplicities of infection) of 0.25, 0.50, 0.75, and 1.0 were tested and a final MOI of 0.50 was chosen due to less dendrite loss as calculated by Neurolucida (S1 Fig). Day 9, 50% of the media was collected, filtered through a 0.8 μm filter, and stored at -80°C for further EV isolation. On Day 11, all 2 mL of the media was collected, filtered, and stored at -80°C until further isolation.

## 4.4. EV isolation

Cyst formation occurs rapidly in neurons following infection. At 3 days post infection the supernatant was collected and EVs isolated through a series of ultracentrifugation methods and a membrane affinity ExoEasy isolation kit [40]. To first remove debris, the supernatant was spun at 500 g for 15 minutes at 4°C. To next remove apoptotic bodies, the supernatant was collected and run on the ultracentrifuge at 15,000 g for 20 minutes at 4°C [140,141]. Lastly, EVs were isolated using the ExoEasy Kit from Qiagen (76064), according to the manufacturer's protocol. Mouse embryo fibroblast EVs from C57BL/6 mice were purchased from STEMON (EDA04–00) to act as a positive EV control. EV isolation and quantification was in line with MISEV guidelines [142]. DOI: https://doi.org/10.17504/protocols.io.n92ldr4eng5b/v1. Collected EVs were counted on the NanoSight NS300 (Malvern Instruments). Isolated EVs were diluted in 1 mL of 1X PBS in a 100X dilution and run on a nanoparticle tracking analysis (NTA). The infusion rate and withdrawal range were both set to 1,000. The screen gain was set to 10 and the detection threshold was set to 5, along with a focus of 60 for all recordings. The NTA provided an average size and concentration of the particles in each sample. Each experiment was performed in technical triplicates. DOI: https://doi.org/10.17504/protocols.io.dm6gp9761vzp/v1.

## 4.5. Transmission Electron Microscopy (TEM)

The presence of EVs was analyzed by TEM. Carbon-coated 200-mesh copper grids with continuous film were glow discharged (EMS 100x) for 20 seconds. To the grids 10 μL of EVs were added until dry. The grids were treated with 30 μL

of 1% uranyl acetate for three different time increments. First for 1 minute, then 30 seconds, and a final 30 seconds. In between each incubation time the uranyl acetate was removed with filter paper before a fresh drop was added. The grids were dried for 10 minutes before imaging. Grids were observed under a Talos L120C TEM (Thermo Scientific) operating at 120 kV. Images were analyzed using ImageJ software.

### 4.6. ELISA

A half-area, high affinity binding 96 well plate was coated with a CD63 capture antibody (Abcam, ab253503) and incubated overnight at 4°C. The plate was then blocked with 1% BSA in 1X PBS for an hour at room temperature. Varying dilutions of sample (1:50, 1:100, and 1:200) were added, along with CD63 standards (Sino Biological, 50557-MNCH), and incubated for 1 hour at 37°C. A Biotinylated-CD63 detection antibody (Abcam, ab253503) was added and incubated for 1 hour at 37°C, protected from light. Streptavidin-HRP was added for 20 minutes, followed by a substrate solution for another 15 minutes. Stop solution was added and the plate was read at 450 nm. Samples were plated in triplicate.

### 4.7. Immunocytochemistry (ICC)

Primary murine cortical neurons were stained 11 days after plating. The supernatant was aspirated, and the cells were washed once with 1X PBS. The cells were fixed with 4% paraformaldehyde (PFA) for 15 minutes rotating, covered from light at room temperature. The solution was aspirated, and coverslips washed three times with 1X PBS. Permeabilization buffer (deionized water supplemented with 10% 10X PBS, 3% BSA, and 0.3% Triton X-100) was added for 10 minutes under rotation, covered from light at room temperature, the solution aspirated, and the coverslips washed three times with 1X PBS. Addition of 5% Donkey Serum (DS), in 1X PBS, was used for a 30 minute block followed by primary antibody ((chicken MAP2 (Abcam, ab5392, 1:1,000), Biotinylated DBA (Vector Labs, B-1035, 1:500), and rabbit NeuN (Abcam, ab177487, 1:500)) diluted in blocking buffer and incubated overnight. Following a further wash, secondary antibody (goat anti-chicken 568 nm (Invitrogen, A11041, 1:1,000), streptavidin conjugate 647 nm (Invitrogen, 532357, 1:1,000), and goat anti-rabbit 488 nm (Invitrogen, A11034, 1:1,000)) was incubated for 1 hour. Following a final wash, slides were cured overnight with VECTASHIELD Mounting Medium with DAPI (Vector Laboratories, H-1500) and imaged with a Leica DMI6000 B inverted fluorescent microscope. When counting the cells "n" represents a single cell throughout multiple experiments.

To visualize EVs in astrocytes, primary murine astrocytes were plated at $2.5 \times 10^5$ cells per well in 2 mL in a 6 well PLL coated plates. PKH67 dye (488 nm) stained the EVs according to the manufacturer's protocol (Sigma, PKH67GL-1KT). EVs were added to astrocytes and incubated at 37°C with 5% $CO_2$ for 2 hours. After incubation, supernatant was aspirated and cells washed once with 1X PBS, as described above. Primary antibody (rat GFAP (Invitrogen, 13-0300, 1:250)) and secondary antibody (donkey anti-rat 568 nm (Invitrogen, A78946, 1:1,000)) diluted in blocking buffer was added. Slides were cured overnight with VECTASHIELD Mounting Medium with DAPI and imaged with a Keyence BZ-X all-in-one fluorescent microscope. DOI: https://doi.org/10.17504/protocols.io.n2bvj9kmwlk5/v1.

For permeabilization experiments following the 2 hour PKH67 labeled EV incubation, astrocytes were further incubated with a primary antibody (rabbit anti-GRA7 (generously gifted by Li-av Zarko, UTMB, 1:250)) diluted in blocking buffer for 1 hour at room temperature. After incubation, supernatant was aspirated and cells washed three times with 1X PBS. For extracellular GRA7 labeling a secondary antibody (donkey anti-rabbit 647 nm (Invitrogen, A31573, 1:1,000)) diluted in blocking buffer was added for 1 hour at room temperature, followed by aspiration and three washes with 1X PBS. Acetone was added to the cells on ice for 10 minutes, followed by aspiration and three washes with 1X PBS. Addition of 5% DS, in 1X PBS, was used to block for 30 minutes at room temperature. The same primary antibody (GRA7) is added and incubated for 1 hour at room temperature, followed by aspiration and three washes with 1X PBS. An additional set of secondary antibodies were added to stain for intracellular GRA7 ((goat anti-rabbit 555 nm (Invitrogen, A21428, 1:1,000))

and (Phalloidin 647 nm (AAT Bioquest, 23127, 1:1,000)) for 1 hour at room temperature, followed by aspiration and three washes with 1X PBS. Slides were cured overnight with VECTASHIELD Mounting Medium with DAPI and imaged with a Keyence BZ-X all-in-one fluorescent microscope.

For blocking experiments endocytic pathway blockers were added 1 hour prior to the addition of PKH67 stained EVs. Astrocytes were incubated with both 25 μM EIPA (5-(N-Ethyl-Nisopropyl) amiloride) (Bio-techne, 3378/10) and 10 μM Cytochalasin D (Gibco, PHZ1063) to observe the EV uptake mechanism. Slides were imaged with a Leica DMI6000 B inverted fluorescent microscope.

### 4.8. Liquid Chromatography (LC)-Mass Spectrometry (MS)/MS

EVs were lysed in RIPA buffer (Thermo, 89901) containing protease inhibitors (Protease Inhibitor Mini Tablets, Thermo, A32955) at 400 μL of RIPA to 100 μL of EVs and incubated on ice for 15 minutes with occasional vortexing. EVs were passed through a 20-gauge needle 5 times and spun at 14,000 g for 15 minutes at 4°C. The supernatant was collected and transferred to an ultracentrifuge tube and washed with 2–3 mL of 8 M urea added and centrifuged at 4,400 rpm for 30 minutes at room temperature. This wash step was repeated three times. To each sample 20 μM dithiothreitol (DTT) (Bio-rad, 1610611) was added and incubated at 37°C for 1 hour. Following incubation 55 μM indoleacetic acid (IAA) (Sigma-Aldrich, I3750) was added and incubated for 30 minutes in the dark at room temperature. A buffer exchange to 50 mM ammonium bicarbonate (ABC) (Fisher Scientific, 02-002-270) was performed at 4,000 rpm for 20–40 minutes at room temperature. This step was repeated 4 times. To each sample 10 μg of Trypsin (Thermo Scientific, 20233) was added and incubated overnight for 16 hours at room temperature. Peptides were collected following centrifugation at 4,600 rpm for 40 minutes at room temperature and concentrated using a Speed-Vac. The peptides were desalted by a C18 ZipTip column (Thermo Scientific, 87782). The peptide solution was then dried by Speed-Vac and stored at -80°C until further LC-MS/MS analysis. Samples were harvested in triplicate.

For LC-MS/MS analysis, fractions were resuspended in 20 μL water with 0.1% formic acid (TCI America, 64-18-6) and separated by nano-LC followed by analysis by on-line electrospray tandem mass spectrometry. Experiments were performed on an EASY-nLC 1200 system (Thermo Fisher) connected to a quadrupole-Orbitrap mass spectrometer Orbitrap Fusion Tribrid Mass Spectrometry equipped with an EASY-Spray ion source (Thermo Fisher). From each peptide, 5 μL was loaded onto the trap column (Acclaim PepMap 100 C18 HPLC column, 75 μm x 25 cm, Thermo Scientific, 164946) with a flow of 10 μl/min for 3 min and subsequently separated on the analytical column (Acclaim PepMap C18) with a linear gradient, from 3% D to 40% D in 55 min. The column was re-equilibrated at initial conditions for 5 min. The flow rate was maintained at 300 nL/min and column temperature was maintained at 45°C. The electrospray voltage of 2.2 kV versus the inlet of the mass spectrometer was used. The Orbitrap Fusion Tribid Mass Spectrometry was operated in the data-dependent mode to switch automatically between MS and MS/MS acquisition. Survey full-scan MS spectra (m/z 375-1500) were acquired with a mass resolution of 60K, followed by fifteen sequential high energy collisional dissociation (HCD). The AGC target was set to $4.0E^5$, and the maximum injection time was 100 ms. MS/MS acquisition was performed in ion trap with the AGC target set to $3E^4$, and the isolation window at 1.6 m/z. Ions with charge states 2+, 3+, and 4+ were sequentially fragmented by HCD with a normalized collision energy (NCE) of 35%. Fixed first mass was set at 100. In all cases, one micro scan was recorded using a dynamic exclusion of 30 seconds.

LC-MS/MS raw data were processed and analyzed using MaxQuant (v 2.1.4.0) aligned to the TgondiiME49 (Toxo DB, version 68) and mouse (Uniport, ID UP000000589) genome databases. In particular, mass tolerances for precursor and fragment ions were 6 and 10 ppm, respectively, the minimum pep-tide length was 6 amino acids, and the maximum number of missed cleavages for trypsin was 2. LC-MS/MS validation of the peptide sequences for the most significant proteins were demonstrated (S2 Fig).

## 4.9. Western blot

EVs were lysed in RIPA buffer (Thermo, 89901) containing Protease Inhibitor Mini Tablets (Thermo, A32955) at 400 µL of RIPA to 100 µL of EVs and incubated on ice for 15 minutes with occasional vortexing. The EVs were passed through a 20-gauge needle 5 times and spun at 14,000 g for 15 minutes at 4°C. The supernatant was then collected and saved at -80°C. Protein concentration was measured by a Micro BCA Protein Assay Kit (Thermo, 23235) according to the manufacturer's protocol. Equal amounts of proteins (2 µg/mL) were separated by SDS-PAGE for 2 hours and transferred onto 0.2 µm nitrocellulose membranes for an additional 1.5 hours. Membranes were washed with 0.1% PBST (1X PBS + 0.1% Tween20) for 5 minutes, shaking at room temperature. Membranes were blocked in a blocking solution (0.1% PBST + 5% Milk) for 30 minutes at room temperature shaking and then probed with primary antibodies (rat anti-CD9 (Sigma, SAB4700567, 1:500), rabbit anti-HSP90B (Proteintech, 11405-1-AP, 1:500), rabbit anti-Vimentin (Abcam, ab92547, 1:500), rabbit anti-Calnexin (Abcam, ab22595, 1:500), rabbit anti-GRA7 (generously gifted by Li-av Zarko, UTMB, 1:1,000)) overnight at 4°C shaking. After washing three times for 5 min in 0.1% PBST, membranes were incubated with HRP-conjugated secondary antibodies (goat anti-rat HRP conjugate (Cell Signaling, 70775, 1:1,000), goat anti-rabbit HRP conjugate (Bio-Rad, 170-5046, 1:5,000)) for 1 h at room temperature shaking. Membranes were washed five times for 10 minutes in 0.1% PBST. Signal was detected with the ECL Western blotting substrate (BioRad, 170-5061) and imaged on the Gel Doc XR+ System (BioRad). Blots were analyzed using ImageJ software.

Primary astrocyte cultures were plated at 1x10^5 cells/mL in 2 mL in 6-well plates. One day after plating Dexamethasone at 100 nM was added to stimulate GLT-1 expression. One day after Dexamethasone treatment experimental conditions were added: media alone, LPS/IFNγ (100 ng/mL and 100 U/mL), EVs from uninfected neurons (2 µg/mL), or EVs from infected neurons (2 µg/mL) and astrocytes incubated for 24 hours. EV concentration was based on publications and material [112,143]. Cells were washed and lysed in RIPA buffer containing protease inhibitors, incubated on ice for 15 minutes with occasional shaking, followed by cell scraping and passage through a 20-gauge needle. The lysate was then spun at 14,000 g for 15 minutes at 4°C, supernatant collected and stored at -80°C. Protein concentration was measured by a BCA Protein Assay Kit (Thermo, 23250) according to the manufacturer's protocol. Equal amounts of proteins (9 µg/mL) were separated by SDS-PAGE for 2 hours and transferred onto a 0.2 µm nitrocellulose membranes for 18.5 hours. Membranes were incubated in a blocking solution for 30 minutes at room temperature and then probed with primary antibodies (rabbit anti-β-actin (Abcam, AB8227, 1:1,000) and guinea-pig anti-GLT-1 (Millipore Sigma, AB1783, 1:500)) overnight at 4 °C. After washing three times for 5 min in 0.1% PBST, membranes were incubated with HRP-conjugated secondary antibodies (goat anti-rabbit HRP conjugate (BioRad, 170-5046, 1:5,000) and goat anti-guinea pig HRP conjugate (Invitrogen, A18769, 1:5,000)) for 1 h at room temperature. Membranes were washed five times for 10 minutes in 0.1% PBST. Signal was detected with the ECL Western blotting substrate and imaged on the Gel Doc XR+ System. Blots were analyzed using ImageJ software. Additional MOIs (0.25, 0.50, 0.75, and 1.0) were also probed for β-actin and GLT-1 with equal amounts of protein added (7 µg/mL). Different concentrations of EVs (3 µg/mL, 0.50 µg/mL, and 0.10 µg/mL) were also probed for β-actin and GLT-1 with equal amounts of protein added (9 µg/mL).

## 4.10. RNA analysis

EV RNA was isolated from neuronal supernatant after a series of ultracentrifugation methods, as mentioned previously. The EV RNA was then isolated through a membrane affinity exoRNeasy Midi Kit (Qiagen, 77144), according to the manufacturer's protocol. RNA concentration and quality was determined by TapeStation RNA ScreenTape (Agilent Technologies) and High Sensitivity RNA Qubit (Thermo Fisher) at the UCR genomics core. Each condition was done in triplicate to obtain biological replicates.

Primary astrocytes were plated at 1x10^5 cells per well in 2 mL in 6 well plates with the same conditions listed under the Western Blot section. After 24 hours of incubation the wells were washed once with 1X PBS and lysed with 350 µl of Buffer RLT (Qiagen RNeasy Mini Kit, 74104). Cells were harvested and passed through a 20-gauge needle 5 times.

Following lysis, the RNeasy Mini Kit was utilized following the manufacturer's protocol. RNA concentration and quality was determined by TapeStation RNA ScreenTape (Agilent Technologies) at the UCR genomics core. Samples with a RIN (RNA Integrity Number) > 9 were chosen. Each condition was conducted in triplicate to obtain biological replicates.

For EV RNA sequencing library construction, quality control, and small RNA sequencing (RNA-Seq) were performed by the UCR Genomics Core. Libraries were constructed with the Qiagen QIAseq miRNA Library Kit (331502) along with the NEBNext Library Quant kit for Illumina (E7630S) for additional quality control. Samples were sequenced on the Illumina NextSeq2000 platform with SE100 around 100 million reads per sample. Raw data in fastq format was processed through CLC genomics workbench (v24.0.2) to remove UMIs (unique molecular identifiers) and aligned to miRbase (v22) mouse genome. Simultaneously, QC and GC content of the clean reads was calculated. All downstream analyses were based on high quality clean data. Differential expression analysis of biological replicates was performed using the DESeq2 R package (v1.20.0). Feature counts were normalized using the normalization function in the DESeq2 package. miRNAs with a p-value <=0.05 and a fold change (FC) >0, found by DESeq2, were assigned as differentially expressed.

For astrocyte RNA sequencing library construction, quality control, and bulk RNA-Seq were performed by UCSD Genomics Core with 25 ng RNA used for each sample. Libraries were generated using the Illumina Stranded mRNA Prep according to the manufacturer's protocol. Constructed libraries were sequenced on the Illumina NovaSeqXPlus LH00444 (240223_LH00444_0061_B223HCLLT4) platform with PE150 around 25 million reads per sample. Raw data in fastq format was processed through FastQC software (v0.11.9) and clean reads were obtained by the removal of reads containing adapters, poly-N and low-quality through Trim Galore software (v0.6.7). Simultaneously, QC and GC content of the clean reads was calculated. All downstream analyses were based on high quality clean data. Reads were aligned to the mouse genome (Gencode, GRCm39, vM34) using STAR (v2.7.11a) and featureCounts (v1.5.0-p3) was used to count the number of reads mapped to each gene. Differential expression analysis of biological replicates was performed using the DESeq2 R package (v1.20.0). Feature counts were normalized using the normalization function in the DESeq2 package. Genes with a p-value <=0.05 and a fold change (FC) >0, found by DESeq2, were assigned as differentially expressed. Gene Ontology (GO) enrichment analysis of differentially expressed genes was performed by the clusterProfiler R package, in which gene length bias was corrected. GO terms with p-values less than 0.05 were considered significantly enriched by differential expressed genes.

### 4.11. Data analysis

Data was plotted using GraphPad Prism 9.0 software, replicates were analyzed using mean and standard deviation, and statistical significance was determined using an unpaired t-test or one way-ANOVA ($p < 0.05$).

### Supporting information

**S1 Fig. An increase in MOI decreases neuronal dendrite numbers.** For quantifications "n" represents a single cell throughout multiple experiments. (**A**) Images taken three days post infection. DAPI stains the nucleus, MAP2 stains the perikarya and dendrites on neurons, DBA stains the sugars of the cyst wall. In all four different infected MOIs there is overlap of the cyst and neuron. Scale bar at 50 µm. (**B**) Quantification of the percentage of infected neurons. The number of cysts and neuronal nuclei were counted and the percentage of infection was calculated (One-way ANOVA, n (0.25) = 26, n (0.50) = 29, n (0.75) = 49, n (1.0) = 55, 0.25 vs. 0.50 p value = 0.0314, 0.25 vs. 0.75 p value = 0.0005, 0.25 vs. 1.0 p value = <0.0001, 0.50 vs. 0.75 p value = 0.7621, 0.50 vs. 1.0 p value = 0.2045, 0.75 vs. 1.0 p value = 0.6699). (**C**) Quantification of the number of dendrites from neurons using Neurolucida. The number of dendrites for each of the infected neuronal groups was compared to the uninfected control group (One-way ANOVA, n (Uninfected) = 29, n (0.25) = 49, n (0.50) = 31, n (0.75) = 54, n (1.0) = 55, Uninfected vs. 0.25 p value = 0.0014, Uninfected vs. 0.50 p value = 0.0036, Uninfected vs. 0.75 p value = <0.0001, Uninfected vs. 1.0 p value = <0.0001).
(TIF)

**S2 Fig. LC-MS/MS peptide sequence confirmations for three different proteins.** (**A**) Extracted ion chromatography (XIC) for the three different significantly regulated proteins. Vimentin (Vim) is upregulated during infection. Reelin (Reln) is downregulated during infection. GRA7 is found in EVs from infected neurons. (**B**) MS1 (first pass) graphs for the peptide sequences for each of the three different proteins. (**C**) MS2 (second pass) for the peptide sequences along with the y and b ions to confirm sequence.
(TIF)

**S3 Fig. Astrocytes do not take up neuronal EVs through macropinocytosis.** (**A**) Orthogonal view of the z-stacks highlighting astrocyte (GFAP) uptake of EVs (PKH67) in the XZ and YZ planes for both groups. (**B**) Fluorescence microscopy of astrocytes after the addition of uptake blockers and EVs from uninfected and *T. gondii* infected neurons. Scale bar: 20 µm. Quantification of the percentage of astrocytes that contain EVs. "n" represents a single cell across multiple experiments (Unpaired t-test, n (Uninfected Control) = 25, n (Infected Control) = 25, n (Uninfected Cytochalasin D/EIPA) = 33, n (Infected Cytochalasin D/EIPA) = 28, Uninfected Control vs. Uninfected Cytochalasin D/EIPA p = 0.1531, Infected Control vs. Infected Cytochalasin D/EIPA p = 0.8922). (**C**) Not all EVs from infected neurons (PKH67) are GRA7 positive. DAPI stains the nucleus, PKH67 stains the EVs, GRA7 stains the internalized GRA7 proteins, Phalloidin stains the actin filaments of the astrocyte.
(TIFF)

**S4 Fig. PCA plots of astrocyte gene expression following the addition of EVs, excluding the LPS/IFNγ data.** (**A**) PCA of astrocytes after 24 hours with the addition of cell culture media (gray), cell culture media without dexamethasone (green), EVs from uninfected neurons (blue) and EVs from infected neurons (purple). (**B**) PCA plot excluding cell culture media without dexamethasone values. (**C**) Volcano plot and GO analysis of DEGs comparing astrocytes RNA after the addition of media, LPS/IFNγ, and EVs from infected neurons. For the volcano plot anything above -log10(0.05) was considered significant. Upregulated genes are shown in red, downregulated shown in blue, and no change in black. For GO analysis bar graphs to the right of 0 indicate upregulated GO terms and to the left represent downregulated GO terms. Bar graphs are colored accorded to adjusted p value (padj).
(TIF)

**S5 Fig. GLT-1 protein concentrations are not changed based on MOI, nor dependent on EV concentrations.** (**A**) Astrocytes with the addition of EVs from uninfected neurons were used as a control to compare with the remaining sample. β-Actin (~42 kDa) was utilized as a loading control. GLT-1 (~62 kDa) was present in all samples with different intensities. Image of gel was taken 166.2 seconds after exposure. Quantification of Western Blot bands using Fiji (ImageJ). The area of the bands in pixels was measured for both β-Actin and GLT-1. A ratio of the pixels (GLT-1/β-Actin) was taken and plotted (One-way ANOVA, n (EVs-Uninfected) = 2, n (EVs-Infected MOI 0.25) = 3, n (EVs-Infected MOI 0.50) = 3, n (EVs-Infected MOI 0.75) = 3, n (EVs-Infected MOI 1.0) = 3, EVs-Uninfected vs. EVs-Infected MOI 0.25 p value = 0.0006, EVs-Uninfected vs. EVs-Infected MOI 0.50 p value = 0.0003, EVs-Uninfected vs. EVs-Infected MOI 0.75 = < 0.0001, EVs-Uninfected vs. EVs-Infected MOI 1.0 p value = 0.0003). (**B**) Western blot for GLT-1 protein expression after the addition of different EV concentrations from uninfected and infected EVs. β-Actin (~42 kDa) was utilized as a loading control. GLT-1 (~62 kDA) was present in all samples with different intensities. Image of gel was taken 29.9 seconds after exposure. Quantification of Western Blot bands using Fiji (ImageJ). The area of the bands in pixels was measured for both β-Actin and GLT-1. A ratio of the pixels (GLT-1/β-Actin) was taken and plotted (One-way ANOVA, n (EVs-Uninfected 3 µg/mL) = 2, n (EVs-Infected 3 µg/mL) = 2, n (EVs-Uninfected 0.50 µg/mL) = 2, n (EVs-Infected 0.50 µg/mL) = 2, n (EVs-Uninfected 0.10 µg/mL) = 2, n (EVs-Infected 0.10 µg/mL) = 2, Uninfected 3 µg/mL vs. Infected 3 µg/mL p value = 0.0403, Uninfected 3 µg/mL vs. Uninfected 0.50 µg/mL p value = 0.0057, Uninfected 3 µg/mL vs. Infected 0.50 µg/mL p value = 0.0045, Uninfected 3 µg/mL vs. Uninfected 0.10 µg/mL p value = 0.0085, Uninfected 3 µg/mL vs. Infected 0.10

µg/mL p value = 0.0080, Infected 3 µg/mL vs. Uninfected 0.50 µg/mL p value = 0.3896, Infected 3 µg/mL vs. Infected 0.50 µg/mL p value = 0.2853, Infected 3 µg/mL vs. Uninfected 0.10 µg/mL p value = 0.6060, Infected 3 µg/mL vs. Infected 0.10 µg/mL p value = 0.5730, Uninfected 0.50 µg/mL vs. Infected 0.50 µg/mL p value = 0.9995, Uninfected 0.50 µg/mL vs. Uninfected 0.10 µg/mL p value = 0.9957, Uninfected 0.50 µg/mL vs. Infected 0.10 µg/mL p value = 0.9979, Infected 0.50 µg/mL vs. Uninfected 0.10 µg/mL p value = 0.9649, Infected 0.50 µg/mL vs. Infected 0.10 µg/mL p value = 0.9756, Uninfected 0.10 µg/mL vs. Infected 0.10 µg/mL p value = >0.9999).
(TIF)

**S1 Table. List of the top 10 significant proteins that are upregulated and downregulated regarding infection when aligned to the mouse genome.** Upregulated proteins are more involved in a stress or interferon response and structural support. Downregulated proteins are more involved with neuronal growth and migration.
(TIF)

**S2 Table. List of the significantly upregulated proteins following infection when aligned to the Toxoplasma genome.** Upregulated proteins consistent of parasite specific dense granules (GRAs) and matrix antigen proteins (MAGs) involved in cyst formation, along with some housekeeping proteins.
(TIF)

**S3 Table. List of the significant differentially expressed miRNA that are upregulated and downregulated regarding infection.** Upregulated miRNA are more related to an anti-inflammatory response. Downregulated miRNA are more involved in a pro-inflammatory response.
(TIF)

**S4 Table. List of the top 10 significant differentially expressed Genes (DEGs) that are upregulated and downregulated regarding infection.** Upregulated genes are more related to an immune and pro-inflammatory response. Downregulated genes are more involved in cellular communication.
(TIF)

## Acknowledgments

Bioanalyzer, Library preparation, and sequencing services on miRNA provided by the Genomics Core, Institute for Integrative Genome Biology, UC Riverside. Matthew Dickson at the University of California, Riverside, Advanced Microscopy and Microanalysis core is acknowledged for performing TEM on the EVs. Brandon Le at the University of California, Riverside, Genomics Core for assistance with the RNASeq data analysis. Quanqing Zhang at the University of California, Riverside, Proteomics Core for running liquid chromatography/mass spectrometry on the EVs. This publication includes data generated at the UC San Diego IGM Genomics Center utilizing an Illumina X Plus that was purchased with funding from a National Institutes of Health SIG grant (#S10 OD026929) and the use of the High-Performance Computing Center (HPCC) at UC Riverside (NSF#2215705). Particular thanks to John Boothroyd and Li-av Zarko for GRA7 reagents and discussions. All authors would like to thank the fundamental resource that is ToxoDB and hope for its continuation.

## Author contributions

**Conceptualization:** Emma H. Wilson.

**Data curation:** Emily Z. Tabaie, Nala Kachour, Arzu Ulu, Stacey Gomez, Zoe A. Figueroa, Kristina V. Bergersen.

**Formal analysis:** Emily Z. Tabaie.

**Funding acquisition:** Emma H. Wilson.

**Investigation:** Emily Z. Tabaie.

**Methodology:** Emily Z. Tabaie, Ziting Gao, Nala Kachour, Arzu Ulu, Stacey Gomez, Zoe A. Figueroa, Kristina V. Bergersen.

**Project administration:** Emma H. Wilson.

**Resources:** Wenwan Zhong, Emma H. Wilson.

**Software:** Emily Z. Tabaie, Ziting Gao.

**Supervision:** Wenwan Zhong, Emma H. Wilson.

**Validation:** Emily Z. Tabaie.

**Visualization:** Emily Z. Tabaie.

**Writing – original draft:** Emily Z. Tabaie.

**Writing – review & editing:** Emily Z. Tabaie, Emma H. Wilson.

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
