## [Decision Letter · Decision Letter 0]

PPATHOGENS-D-24-02418

Toxoplasma gondii infection of neurons alters the production and content of extracellular vesicles altering astrocyte phenotype and contributing to the loss of GLT-1 in the infected brain

PLOS Pathogens

Dear Dr. Wilson,

Thank you for submitting your manuscript to PLOS Pathogens. After careful consideration, we feel that it has merit but does not fully meet PLOS Pathogens's publication criteria as it currently stands. Therefore, we invite you to submit a revised version of the manuscript that addresses the points raised during the review process.

Please submit your revised manuscript within 60 days Feb 20 2025 11:59PM. If you will need more time than this to complete your revisions, please reply to this message or contact the journal office at plospathogens@plos.org. Please include the following items when submitting your revised manuscript:

We look forward to receiving your revised manuscript.

Kind regards,

Eric Y Denkers

Academic Editor

PLOS Pathogens

Tracey Lamb

Section Editor

PLOS Pathogens

 Sumita Bhaduri-McIntosh

Editor-in-Chief

PLOS Pathogens

orcid.org/0000-0003-2946-9497 

Michael Malim

Editor-in-Chief

PLOS Pathogens

orcid.org/0000-0002-7699-2064

**Additional Editor Comments :**

The reviewers find this work to be of interest but identify several weaknesses and areas that could be improved with additional data. An significant weakness is that conclusions are based solely on in vitro generated data that make up this study. Linking these findings to events occurring in the brain during infection may be inherently difficult, but reviewer 3 suggests some approaches.

**Journal Requirements:**

At this stage, the following Authors/Authors require contributions: Emily Z Tabaie, Ziting Gao, Stacey Gomez, Kristina V Bergersen, Wenwan Zhong, and Emma H. Wilson. Please ensure that the full contributions of each author are acknowledged in the "Add/Edit/Remove Authors" section of our submission form.

https://journals.plos.org/plospathogens/s/submission-guidelines#loc-parts-of-a-submission

4) We do not publish any copyright or trademark symbols that usually accompany proprietary names, eg ©,  ®, or TM  (e.g. next to drug or reagent names). Therefore please remove all instances of trademark/copyright symbols throughout the text, including:

- ® on pages: 8, and 9

- TM on page: 10.

5) Please upload all main figures as separate Figure files in .tif or .eps format. For more information about how to convert and format your figure files please see our guidelines: 

6) Please upload a copy of Figure 5E which you refer to in your text on page 19. Or, if the figure is no longer to be included as part of the submission please remove all reference to it within the text.

7) We have noticed that you have uploaded Supporting Information files, but you have not included a list of legends. Please add a full list of legends for your Supporting Information files after the references list. Please also include the labels of the supplementary tables.

8) Thank you for stating that "Raw sequencing and processing files for RNA and miRNA Sequencing have been deposited in the GEO database under accession code: PRJNA1158965. Raw and processed files for LC-MS/MS protein analysis have been deposited in the MassIVE database under accession code: MSV000096222." 

We strongly recommend all authors decide on a data sharing plan before acceptance, as the process can be lengthy and hold up publication timelines. Please note that, though access restrictions are acceptable now, your entire data will need to be made freely accessible if your manuscript is accepted for publication. This policy applies to all data except where public deposition would breach compliance with the protocol approved by your research ethics board. If you are unable to adhere to our open data policy, please kindly revise your statement to explain your reasoning and we will seek the editor's input on an exemption. Please be assured that, once you have provided your new statement, the assessment of your exemption will not hold up the peer review process. 

9) Please amend your detailed Financial Disclosure statement. This is published with the article. It must therefore be completed in full sentences and contain the exact wording you wish to be published.

1) State the initials for these grants " (NSF#2215705) and (S10 OD026929), alongside each funding source, of each author to receive each grant . For example: "This work was supported by the National Institutes of Health (####### to AM; ###### to CJ) and the National Science Foundation (###### to AM)." 

2) State what role the funders took in the study. If the funders had no role in your study, please state: "The funders had no role in study design, data collection and analysis, decision to publish, or preparation of the manuscript.".

10) Your current Financial Disclosure states, " This work was supported, in part, by funding from the National Institute of Health NS137603 and AI158417 to EHW and funds from the Graduate Division at the University of California, Riverside to EHW and from the Division of Biomedical Sciences, UCR to EZT. Support for core equipment used in this study was provided through the National Sceince Foundation (NSF#2215705) and the National Institutes of Health (S10 OD026929).".

However, your funding information on the submission form indicates only two funders. Please ensure that the funders and grant numbers match between the Financial Disclosure field and the Funding Information tab in your submission form. Note that the funders must be provided in the same order in both places as well.

Please indicate by return email the full and correct funding information for your study and confirm the order in which funding contributions should appear. Please be sure to indicate whether the funders played any role in the study design, data collection and analysis, decision to publish, or preparation of the manuscript.

**Reviewers' Comments:**

Reviewer's Responses to Questions

**Part I - Summary**

Reviewer #1: The authors seek to explore how Toxoplasma infection influences the responses or characteristics of cells in the vicinity of the infected cell that may not themselves be infected. These studies were informed by previous studies that had found that secretions from neurons including secretion of extracellular vesicles (EVs) can modulate gene expression of astrocytes. The approach of the authors was to first characterize the effects of Toxoplasma infection on EVs that are released from Toxoplasma infected neurons as compared to EVs from uninfected neurons. Specifically, they assess the effects on infection on numbers of EVs released as well as on the content of their cargo (protein and nucleic acid (miRNA) content). They then proceed to evaluate the effects of these EVs on astrocytes by incubating EVs with astrocytes, after which they evaluate the gene expression of the astrocytes. They observed that EVs from Toxoplasma infected neurons have a modest effect on GLT protein levels in astrocytes through an uncertain mechanism. Overall, the findings are interesting and significant. However, below are suggestions that they should consider adding to their experimental approaches that could better clarify if not strength their findings.

Reviewer #2: In this study, Tabaie and colleagues characterize how infection of neurons with T. gondii cysts alters the quality of the extracellular vesicles released by neurons and look at their effects on astrocytes. The study addresses an important issue regarding how neuronal-astrocyte interactions may be altered, especially focusing on glutamate receptor expression on astrocytes, which may have an impact on glutamate homeostasis. The experimental approaches are primarily centered on in vitro cultures and modelling the interactions by co-culturing of neuron-derived EVs and primary astrocyte cultures. The study came up with potentially interesting observations, which need to be further validated. Overall, the study is still in a descriptive stage, but this may be of interest to the readership.

Reviewer #3: The manuscript from Tabaie, et al explores the impact of T. gondii infection of neurons on extracellular vesicles in vitro. The authors infect primary cortical neurons with T. gondii and examine the number and content of EVs derived from the neuronal cultures. They found that infected neurons produced fewer EVs and the EVs contain neuronal and parasite proteins. The neuronal EVs were then used to treat cultured astrocytes. The authors observed that astrocytes responded to the EVs from the infected cells and induced genes related to an immune response. Overall, the results suggest that infected neurons are altered and their EV content could impact glial function in the brain. The results may reflect processes in the infected brain, but this was not confirmed in the current manuscript.

**Part II – Major Issues: Key Experiments Required for Acceptance**

Reviewer #1: The evaluation of EVs from the neuronal primary cells must be challenging. As they report, neuronal cells are sensitive to the infection loaded, which necessitated that they perform experiments with a relatively low parasite : neuronal cell ratio. It would be informative to include the cell numbers (neurons) at the start of infection as it provides necessary information about the system. The authors should consider assessing the effect of inert particle uptake by these neuronal cells. If inert particles do not affect EV particle release, it should support their observation that the effect of Toxoplasma is physiologically distinct.

They show the size of particles obtained from TEM. Although it is often used in the characterization of EVs, the TEM protocol results in dehydration of the samples, which results in significant shrinkage of EVs. They show that the average size from uninfected and infected samples is approximately 75 – 77nm. So, while the morphology might be informative, the size obtained from TEM isn’t.

The point just raised is amplified by the ‘noisy’ NTA profile of EVs in their preparations. Based on the NTA profiles, there appears to be a significant shift towards much smaller particles after Toxoplasma infection. There is ample information that EV isolation kits (protocols) can differentially enrich or exclude subpopulations of EVs. While CD63 is an accepted reproducible component of EVs, it would be informative if they performed Western blots where other accepted EV markers (TSG101, CD9 and others) are evaluated. Such analysis could provide a bit more insight on the underlying mechanism of Toxoplasma suppression of EV release.

The LC/MS/MS data in Figure 3 is fine. However, it is useful to provide validation of some kind that supports that those molecules are indeed present in their preparation. A Western blot showing a representative few of these identified molecules would suffice.

In the experiments in Figure 5, z series images are used to assert that EVs from neuronal cells are taken up by astrocytes. A more quantifiable approach would be to perform immunofluorescence to detect a known EV component (parasite derived molecule, for example) under permeabilized or non-permeabilized conditions. The compartment in which the molecule of interest is localized would less ambiguous.

In the experiments described in Figure 6, the PCA plots include gene expression after LPS/IFNy. There is no mention whether EVs from infected neurons impact the activation by LPS/IFNy

Reviewer #2: 1. As written, some sections of the Abstract may be misleading. For example, “EVs from infected neurons contained secreted Toxoplasma proteins GRA1, GRA2, GRA7, MAG1 and MAG2 associated with cyst formation”---- not all the proteins enumerated are necessarily cyst-specific. Also, “EVs from infected neurons altered gene expression--- leading to a downregulation of GLT-1 protein expression..”. Here the word “leading” is not supported by the data, since we do not know how the EVs decreased GLT-1 protein but not mRNA levels.

2. The most interesting part of the data set presented is in Figure 6, showing an IFN-like signature induced by EVs from infected neurons and decreased GLT-1 protein expression, but no suppression of its mRNA. There is also a group stimulated with LPS/IFNg, but this is not shown for the rest of the Figure. It would be important to show whether this strong stimulation combination also exhibits the same properties as the EVs from infected neurons, perhaps in magnified way. Importantly, it would be important to probe, whether blockade of interferon-signaling is required to observe the downregulation of GLT-1 protein. A functional assay to show the effect of the downregulation would be good to also include.

Reviewer #3: 1) To make a potential connection between the observations made in vitro to the brain during infection, the gene expression changes in the astrocytes in vitro could be made to the known changes in astrocyte gene expression in the brain in response to infection. The authors nicely tie their results back to GLT-1, but are the other genes reflected in vivo (response to virus, etc.)

2) A major conclusion is that infected neurons generate fewer EVs. While the authors confirm that there is a similar number of neurons in each culture, the neurons have an altered morphology and appear to be sicker, including condensed nuclei. Additional measures of neuronal health would be helpful in understanding the health of the neuron. The authors state that higher MOIs “overwhelm the cultures” and lead to neuronal damage. At the lower MOI, the cells may still be quite sick.

3) In the PCA plot examining changes in gene expression in astrocytes, the authors compare the changes to cells treated with LPS and IFN-g. The inclusion of this treatment group was incredible useful to gauge the degree of the response to the EVs. The zoom in on the PCA plot is necessary to appreciate differences between the EV-treated groups. The authors should comment that the PCA plot does not show dramatic changes between the groups. The samples could also be re-analyzed without the LPS/IFN group to make the differences more obvious (or not).

4) In several panels including Figure 1B, 1C, 5C-D, it is unclear what the “n” represents. For example, is each data point in the graph a single cell, single culture, or single experiment (with data averaged across wells). Clarity in the figure legend would be helpful to understand where the variation is coming from.

**Part III – Minor Issues: Editorial and Data Presentation Modifications**

Reviewer #1: Minor:

Figure 4 legend needs to be clarified ‘miRNAs after the addition of EVs from Uninfected EV and Infected EVs’. Were these EVs recovered from infected cells or were they added to infected cells?

Reviewer #2: (No Response)

Reviewer #3: 5) In the first line of the abstract, the authors state that neurons are the only cell type where cysts form, which is not entirely accurate. Later in the manuscript the authors state that the majority of cyst formation is in neurons. The abstract should be amended to be consistent with the latter statement. Also, the infection is likely not dormant for the lifetime of the host (particularly in mice).

6) Line 62-63 of the introduction needs a citation at a minimum. The statement that the parasite manipulates cell signaling pathways, inhibits host defense, etc. is all true but perhaps based on multiple cell types. Far less is known about how the parasite specifically affects neurons. The statement should be supported by citations and the cells types in which these findings were observed should be reflected.

7) Line 76. Could astrocyte-neuronal communication also be affected by secreted factors and not just secreted EVs?

8) Figure 4B would be improved by including the miRNA in the heatmap.

PLOS authors have the option to publish the peer review history of their article (what does this mean? ). If published, this will include your full peer review and any attached files.

**Do you want your identity to be public for this peer review?** For information about this choice, including consent withdrawal, please see our Privacy Policy .

Reviewer #1: No

Reviewer #2: No

Reviewer #3: No

**Figure resubmission:**
---

## [Decision Letter · Decision Letter 1]

Dear Dr. Wilson,

We are pleased to inform you that your manuscript 'Toxoplasma gondii infection of neurons alters the production and content of extracellular vesicles directing astrocyte phenotype and contributing to the loss of GLT-1 in the infected brain' has been provisionally accepted for publication in PLOS Pathogens.

Best regards,

Eric Y. Denkers

Academic Editor

PLOS Pathogens

Tracey Lamb

Section Editor

PLOS Pathogens

Sumita Bhaduri-McIntosh

Editor-in-Chief

PLOS Pathogens

orcid.org/0000-0003-2946-9497

Michael Malim

Editor-in-Chief

PLOS Pathogens

orcid.org/0000-0002-7699-2064

The manuscript has been significantly strengthened and the reviewers unanimously recommend acceptance. Congratulations!

Reviewer Comments (if any, and for reference):

Reviewer's Responses to Questions

**Part I - Summary**

Reviewer #1: Authors were responsive to reviewer suggestions. Manuscript is much improved. The complex interactions associated with Toxoplasma infection of neurons are brought to the fore.

Reviewer #2: The authors have addressed the concerns and questions raised in the prior review. The revised manuscript is more complete.

Reviewer #3: The authors have adequately addressed all of the concerns raised upon the initial review.

**Part II – Major Issues: Key Experiments Required for Acceptance**

Reviewer #1: No new concerns from this reviewer.

Reviewer #2: (No Response)

Reviewer #3: (No Response)

**Part III – Minor Issues: Editorial and Data Presentation Modifications**

Reviewer #1: No new concerns from this reviewer.

Reviewer #2: (No Response)

Reviewer #3: (No Response)

PLOS authors have the option to publish the peer review history of their article (what does this mean? ). If published, this will include your full peer review and any attached files.

**Do you want your identity to be public for this peer review?** For information about this choice, including consent withdrawal, please see our Privacy Policy .

Reviewer #1: No

Reviewer #2: No

Reviewer #3: No

---

## [Editor Report · Acceptance letter]

Dear Professor Wilson,

We are delighted to inform you that your manuscript, "Toxoplasma gondii infection of neurons alters the production and content of extracellular vesicles directing astrocyte phenotype and contributing to the loss of GLT-1 in the infected brain," has been formally accepted for publication in PLOS Pathogens.

Best regards,

Sumita Bhaduri-McIntosh

Editor-in-Chief

PLOS Pathogens

orcid.org/0000-0003-2946-9497

Michael Malim

Editor-in-Chief

PLOS Pathogens

orcid.org/0000-0002-7699-2064